EMBO
Molecular Medicine

# Comprehensive evaluation of coding region point mutations in microsatellite-unstable colorectal cancer

Johanna Kondelin[1,2], Kari Salokas[3,4], Lilli Saarinen[2], Kristian Ovaska[2], Heli Rauanheimo[1,2], Roosa-Maria Plaketti[1,2], Jiri Hamberg[1,2], Xiaonan Liu[3,4] (iD), Leena Yadav[3,4], Alexandra E Gylfe[1,2], Tatiana Cajuso[1,2], Ulrika A Hänninen[1,2], Kimmo Palin[1,2], Heikki Ristolainen[1,2], Riku Katainen[1,2], Eevi Kaasinen[1,2], Tomas Tanskanen[1,2], Mervi Aavikko[1,2], Minna Taipale[5], Jussi Taipale[1,2,6,7] (iD), Laura Renkonen-Sinisalo[8], Anna Lepistö[8], Selja Koskensalo[9], Jan Böhm[10], Jukka-Pekka Mecklin[11,12], Halit Ongen[13,14,15], Emmanouil T Dermitzakis[13,14,15], Outi Kilpivaara[1,2], Pia Vahteristo[1,2], Mikko Turunen[2], Sampsa Hautaniemi[2], Sari Tuupanen[1,2], Auli Karhu[1,2], Niko Välimäki[1,2], Markku Varjosalo[3,4], Esa Pitkänen[1,2] (iD) & Lauri A Aaltonen[1,2,*] (iD)

## Abstract

Microsatellite instability (MSI) leads to accumulation of an excessive number of mutations in the genome, mostly small insertions and deletions. MSI colorectal cancers (CRCs), however, also contain more point mutations than microsatellite-stable (MSS) tumors, yet they have not been as comprehensively studied. To identify candidate driver genes affected by point mutations in MSI CRC, we ranked genes based on mutation significance while correcting for replication timing and gene expression utilizing an algorithm, MutSigCV. Somatic point mutation data from the exome kit-targeted area from 24 exome-sequenced sporadic MSI CRCs and respective normals, and 12 whole-genome-sequenced sporadic MSI CRCs and respective normals were utilized. The top 73 genes were validated in 93 additional MSI CRCs. The MutSigCV ranking identified several well-established MSI CRC driver genes and provided additional evidence for previously proposed CRC candidate genes as well as shortlisted genes that have to our knowledge not been linked to CRC before. Two genes, *SMARCB1* and *STK38L*, were also functionally scrutinized, providing evidence of a tumorigenic role, for *SMARCB1* mutations in particular.

**Keywords** cancer genetics; colorectal cancer; microsatellite instability

**Subject Categories** Cancer; Chromatin, Epigenetics, Genomics & Functional Genomics; Systems Medicine

## Introduction

Colorectal cancer (CRC) is one of the most fatal cancers in Western countries leading to death in nearly 50% of the cases (Jemal *et al*, 2011). Approximately 15% of CRCs exhibit microsatellite instability (MSI), which results from defective DNA mismatch repair (MMR) machinery (Boland & Goel, 2010). This is most often the result of

1 Medicum/Department of Medical and Clinical Genetics, University of Helsinki, Helsinki, Finland
2 Genome-Scale Biology Research Program, Research Programs Unit, University of Helsinki, Helsinki, Finland
3 Institute of Biotechnology, University of Helsinki, Helsinki, Finland
4 Helsinki Institute of Life Science, University of Helsinki, Helsinki, Finland
5 Division of Functional Genomics, Department of Medical Biochemistry and Biophysics (MBB), Karolinska Institutet, Stockholm, Sweden
6 Department of Biosciences and Nutrition, Karolinska Institutet, Huddinge, Sweden
7 Science for Life Center, Huddinge, Sweden
8 Department of Surgery, Helsinki University Central Hospital, Hospital District of Helsinki and Uusimaa, Helsinki, Finland
9 The HUCH Gastrointestinal Clinic, Helsinki University Central Hospital, Helsinki, Finland
10 Department of Pathology, Jyväskylä Central Hospital, Jyväskylä, Finland
11 Department of Surgery, Jyväskylä Central Hospital, University of Eastern Finland, Jyväskylä, Finland
12 Department Sport and Health Sciences, University of Jyväskylä, Jyväskylä, Finland
13 Department of Genetic Medicine and Development, University of Geneva Medical School, Geneva, Switzerland
14 Institute for Genetics and Genomics in Geneva (iGE3), University of Geneva, Geneva, Switzerland
15 Swiss Institute of Bioinformatics, Geneva, Switzerland
*Corresponding author. Tel: +358 2941 25595; E-mail: lauri.aaltonen@helsinki.fi

hypermethylation of the promoter of *MLH1*, one of the central genes involved in MMR.

It is estimated that approximately 90% of CRCs are sporadic, whereas the remaining 10% arise due to inherited predisposition (Bogaert & Prenen, 2014). The most common form of inherited predisposition is Lynch syndrome, where the individual inherits a germline mutation in one of the MMR genes (*MLH1, MSH2, MSH6, PMS2*) and is therefore highly predisposed to CRC and endometrial cancer (Boland & Goel, 2010). In addition to CRC, MSI is also observed in approximately 15% of sporadic endometrial and gastric cancers (Hamelin *et al*, 2008). MSI CRCs arise through a distinct genetic pathway as compared to microsatellite-stable (MSS) CRCs (Boland & Goel, 2010). The defective MMR machinery results in the accumulation of an excessive number of mutations in the genome. Most of the mutations are small insertions and deletions (indels) that target short nucleotide repeats, microsatellites. Genes that provide growth advantage to cells via loss-of-function mutations in microsatellites, or MSI target genes (Duval & Hamelin, 2002), have been extensively studied and numerous genes have been published as candidate targets and thus putative tumor suppressors (Alhopuro *et al*, 2012; Kondelin *et al*, 2017). MSI tumors also contain an order of magnitude more point mutations than MSS tumors (Boland & Goel, 2010), yet to date the point mutations in MSI CRCs have been mostly overlooked. Only few genes with causative point mutations have been identified in this tumor type. Most of these have been flagged by missense mutation hot spots (e.g., *BRAF, KRAS, CTNNB1,* and *PIK3CA*), a mutation pattern typical of oncogenes (Fearon, 2011).

In our past efforts, we have identified candidate oncogenes with missense mutation hot spots based on next-generation sequencing (NGS) data from a small discovery set of MSI CRCs (Gylfe *et al*, 2013; Tuupanen *et al*, 2014). To our knowledge, however, only few studies have attempted to systematically characterize the full landscape of coding point mutations in MSI CRC in order to identify new driver genes (Cancer Genome Atlas Network 2012; Seshagiri *et al*, 2012; Kim *et al*, 2013; Cortes-Ciriano *et al*, 2017).

In the past few years, NGS has been largely accepted into both research and clinical use, and numerous mutations—both somatic and germline—have been reported to contribute to disease. There is, however, debate on which of the genes and mutations reported are truly significant for disease (Gonzalez-Perez *et al*, 2013). The distinction between driver genes and the incidentally mutated passengers is a challenge that is augmented in MSI tumors due to their high mutation load. It is, however, established that mutation frequency solely is a poor predictor of causality (Vogelstein *et al*, 2013). In attempt to account for other important factors, algorithms have been developed to predict which genes are likely cancer-driving genes based on several parameters (Lawrence *et al*, 2013; Mularoni *et al*, 2016).

In this study, we utilized a discovery set of 24 exome-sequenced sporadic MSI CRCs and respective normals, and 12 whole-genome-sequenced sporadic MSI CRCs and respective normals to identify driver genes affected by point mutations in MSI CRC (Fig 1). The top 73 genes predicted as the most likely to be causative were re-sequenced in a validation set of 93 additional MSI CRCs. From this effort, *SMARCB1* emerged as our top candidate for a novel MSI CRC driver gene.

To continue on our previous studies where candidate MSI CRC oncogenes were identified based on mutation hot spots in a smaller dataset (Gylfe *et al*, 2013; Tuupanen *et al*, 2014), we repeated the hot spot analysis in this dataset of 36 exome- or whole-genome-sequenced sporadic MSI CRCs and corresponding normals. Hence, genes containing mutation hot spots in these somatic point mutation data were detected (Figs 1 and 2). From this set of hot spots, the 90 novel hot spots as well as seven previously studied hot spots were re-sequenced in the validation set of 93 additional MSI CRCs. From this effort, seven new candidate oncogenes emerged (*CORIN, KLHL6, PCDHB16, PLEKHG1, PROS1, SPP2,* and *TROAP*). To our knowledge, this study represents the first effort to uncover driver point mutations in MSI CRC utilizing deep sequencing of a large set of tumors for validation.

## Results

In order to identify new candidates for driver genes affected by point mutations in MSI CRC, we analyzed sequencing data from a discovery set of 36 exome- or whole-genome-sequenced MSI CRCs and respective normals. MutSigCV analysis was performed on the somatic single-nucleotide variation (SNV) data to identify the genes most likely to display an excess of point mutations due to selection, and the resulting top 73 genes were further validated by MiSeq sequencing in a validation set of 93 additional MSI CRCs. Next, a new algorithm, OncodriveFML, had become available during the study and was utilized on the somatic SNV data from the MiSeq sequencing to identify the most likely candidates for previously unknown CRC-driving genes. Of these, *SMARCB1* and *STK38L* were further validated in functional experiments. The analysis workflow is summarized in Fig 1.

In addition, to continue on our previous efforts (Gylfe *et al*, 2013; Tuupanen *et al*, 2014), we performed an analysis on genes containing somatic mutation hot spots—mutations residing in either the same or two adjacent codons, or two bases flanking an exon–intron boundary—in at least two samples. Ninety-seven hot spots from 94 genes were selected for further validation in the set of 93 additional MSI CRCs. The analysis workflow is summarized in Fig 2.

### Characterization of the SNVs in the discovery set of 36 exome- or whole-genome-sequenced MSI CRCs

A median of 778 somatic SNVs were found in the exome kit-targeted region of the 36 NGS samples. On average, 75% of the targeted bases had a coverage of $\geq 21$ reads, and the average coverage of the targeted regions was 47. The mean frequencies of the SNV types are shown in Appendix Fig S1. The most frequent mutation type was C:G>T:A (54.5%) as was to be expected in MSI CRC (Alexandrov *et al*, 2013; Tuupanen *et al*, 2014). The mutation frequencies per sample are shown in Appendix Fig S2.

### MutSigCV yields a ranking of genes based on the discovery set

MutSigCV was run on the somatic SNV data from the discovery set of 36 exome- or whole-genome-sequenced MSI CRCs, and yielded a ranking of 7,511 genes (Dataset EV1, Fig 1). Genes with mutations

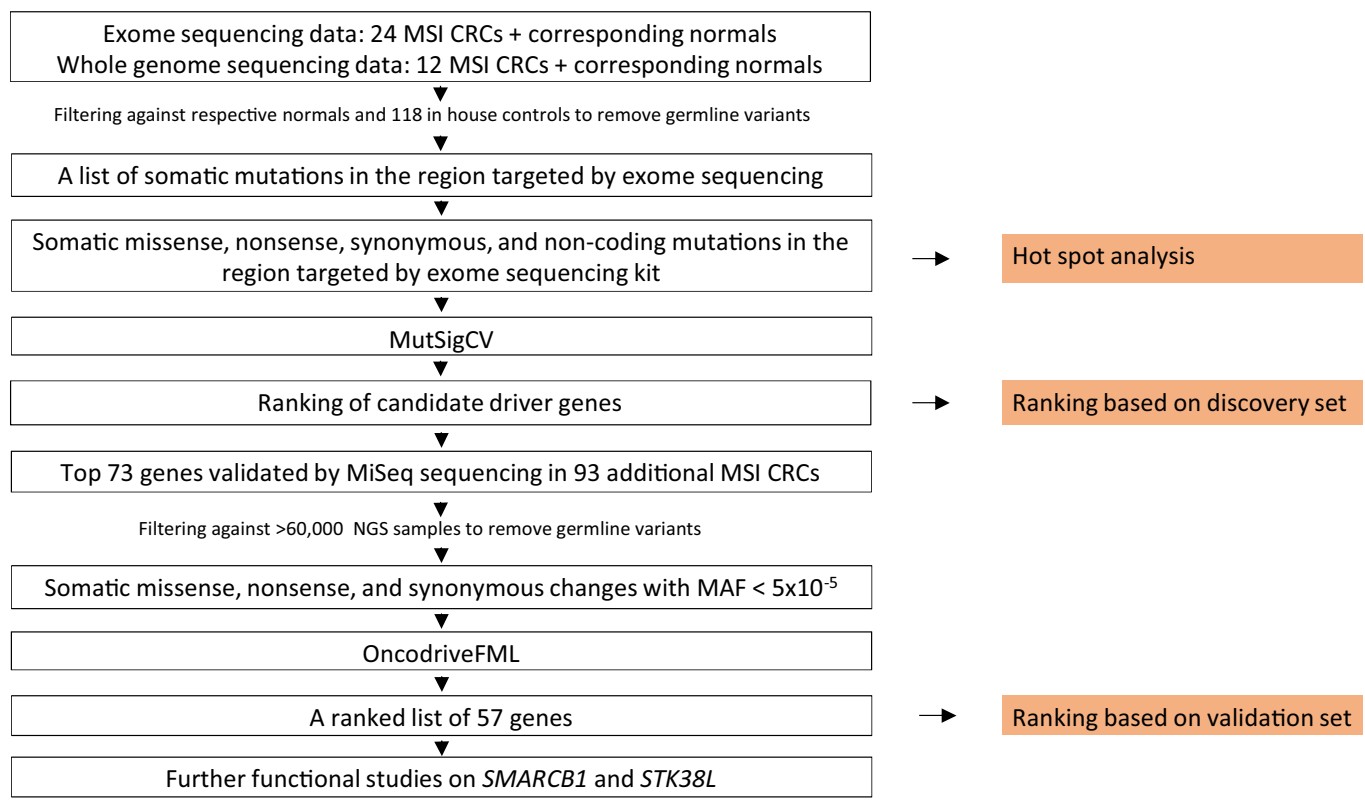

**Figure 1.  Schematic representation of the overall study design.**

We utilized exome sequencing data from 24 MSI CRCs and corresponding normals and whole-genome sequencing data from 12 MSI CRCs and corresponding normals. The tumor data were filtered against the corresponding normals as well as outside controls to remove germline variants. Only the areas targeted by the exome sequencing kit were included in the analysis. Insertions and deletions were then filtered out, leaving the somatic missense, nonsense, synonymous, and noncoding mutations in the region targeted by the exome sequencing kit. On this set of variants, (i) hot spot analysis was performed, and (ii) MutSigCV was utilized, resulting in a ranking of genes. Genes with mutations in one or two tumors only were excluded. The top 73 genes were validated by MiSeq sequencing in a validation set of 93 additional MSI CRCs. The data were filtered against > 60,000 outside controls to remove germline variants. Insertions and deletions were left out, resulting in a set of somatic missense, nonsense, and synonymous changes with MAF < $5 \times 10^{-5}$. On this set of variants, OncodriveFML was utilized. Genes with mutations in only one tumor were excluded, resulting in a ranking of 57 candidate driver genes. Further functional studied were carried out for *SMARCB1* and *STK38L*.

in only one or two tumors were excluded (resulting in a ranked list of genes with non-synonymous or splice site mutations in at least three tumors; "Ranking based on discovery set"; Fig 1, Dataset EV2). The genes did not display statistically significant *P*-values (Dataset EV1), which is likely due to the small size of the discovery set. Therefore, MutSigCV was rather used as a ranking tool. Among the top ten genes, three well-established MSI CRC driver genes (*BRAF*, *CTNNB1*, and *PIK3CA*) were found (Dataset EV2) (Shitoh *et al*, 2001; Davies *et al*, 2002; Fearon, 2011), providing confidence in the MutSigCV approach to identify MSI CRC driver genes.

**Characterization of the SNV mutations in the validation set of 93 MiSeq-sequenced MSI CRCs**

From the ranking based on the discovery set, the coding regions of the top 73 genes were targeted for further validation in MiSeq sequencing of the validation set of 93 MSI CRCs (Dataset EV2). A median of six somatic SNVs were found in the 73 MutSigCV-ranked genes. On average, 92% of the targeted bases had a coverage of ≥ 21 reads, and the average coverage of the targeted regions was 222. As with the 36 exome- or genome-sequenced tumors, the most frequent somatic mutation type was C:G>T:A (47.8%)

(Appendix Fig S3). The mutation frequencies per sample are shown in Appendix Fig S4.

**OncodriveFML analysis reveals novel candidate MSI CRC driver genes**

OncodriveFML, suitable for ranking genes in smaller datasets, was run on the somatic SNV data from the 73 MiSeq-sequenced genes from the validation set of 93 MSI CRCs (Dataset EV3) (Mularoni *et al*, 2016). From the resulting ranking (OncodriveFML ranking; Dataset EV4), genes with only one mutation were excluded as OncodriveFML does not calculate *q*-values for them. The resulting data were therefore a ranking of 57 genes ("Ranking based on validation set"; Fig 1, Table 1, Final ranking; Dataset EV4). On top of the ranking, there were eight genes (*BRAF*, *CTNNB1*, *CASP8*, *CCDC47*, *STK38L*, *ENO3*, *PIK3CA*, and *SMARCB1*) with a *q*-value smaller than 0.1. PolyPhen and SIFT predictions for the variations in these eight genes are featured in Dataset EV3.

Three of the genes found among the top genes (*BRAF*, *CTNNB1*, and *PIK3CA*) are previously well-established oncogenic drivers of MSI CRC (Shitoh *et al*, 2001; Davies *et al*, 2002; Fearon, 2011). Characteristic of oncogenes, they each harbor one

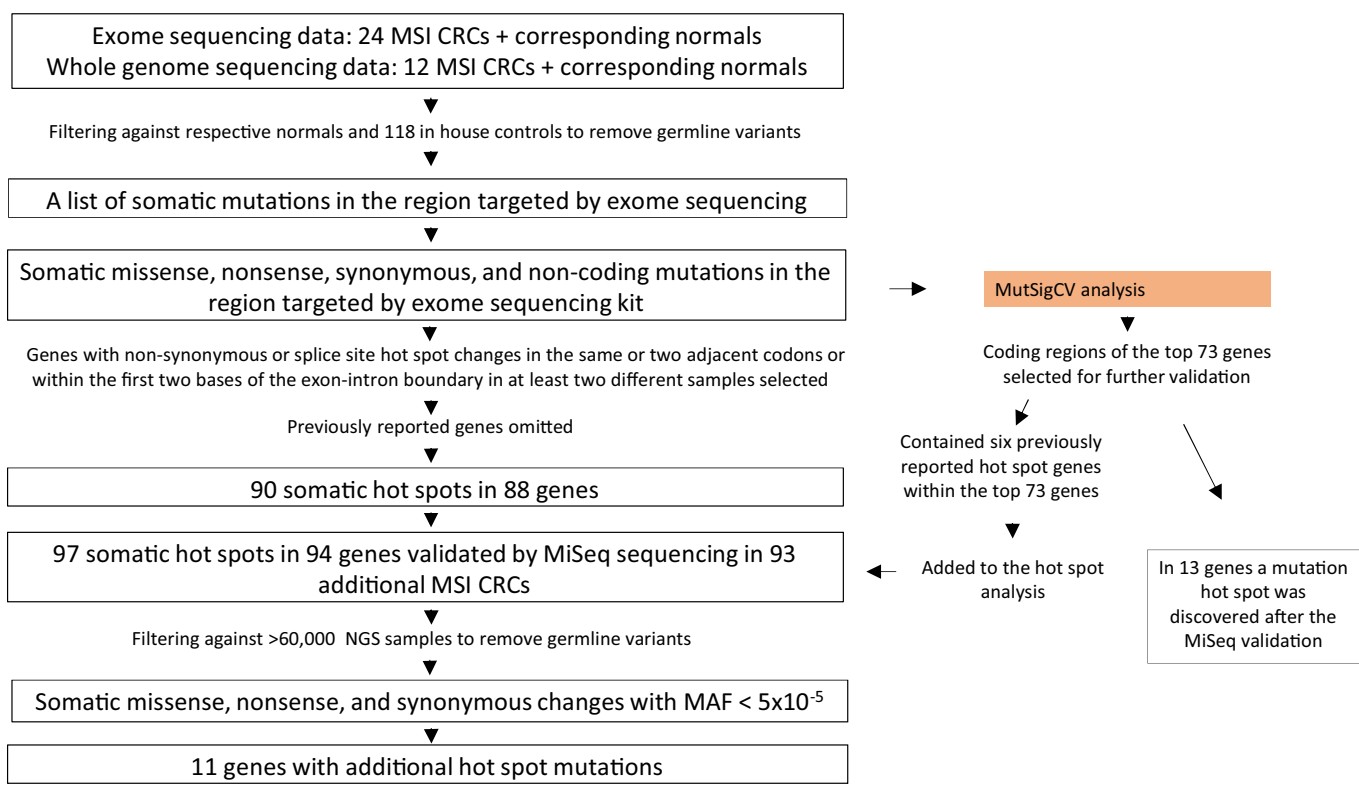

**Figure 2. Schematic representation of the hot spot analysis.**

We utilized exome sequencing data from 24 MSI CRCs and corresponding normals and whole-genome sequencing data from 12 MSI CRCs and corresponding normals. The tumor data were filtered against the corresponding normals as well as outside controls to remove germline variants. Only the areas targeted by the exome sequencing kit were included in the analysis. Insertions and deletions were then filtered out, leaving the somatic missense, nonsense, synonymous, and noncoding mutations in the region targeted by the exome sequencing kit. On this set of variants, (i) hot spot analysis was performed, and (ii) MutSigCV was utilized. In the hot spot analysis, genes with non-synonymous or splice site hot spot changes—mutations residing in either the same or two adjacent codons, or two bases flanking an exon–intron boundary—were identified. Previously reported hot spot genes were omitted. The resulting list consisted of 90 somatic hot spots in 88 genes. The MutSigCV analysis included six previously reported hot spot genes, which were then added to the hot spot list. The hot spot list entering validation therefore consisted of 97 somatic hot spots in 94 genes, and these 97 hot spots were re-sequenced with MiSeq sequencing in the validation set of 93 additional MSI CRCs. The data were filtered against > 60,000 outside controls to remove germline variants. Insertions and deletions were left out, resulting in a set of somatic missense, nonsense, and synonymous changes with MAF < 5 × 10$^{-5}$. In the MiSeq data, 11 of the 94 genes contained additional hot spot mutations.

or more non-synonymous missense mutation hot spots (Dataset EV3).

*CASP8* has been listed as significantly mutated in hypermutable CRCs (Cancer Genome Atlas Network, 2012) and suggested to be a CRC suppressor gene (Kim *et al*, 2003). In our data, we observed ten non-synonymous changes in *CASP8*, of which all are found within functional domains of the gene (Pfam domains, Dataset EV3; Finn *et al*, 2016). Three of the 10 variants are in the death effector domains. Of the ten variants, eight were predicted damaging by both SIFT and PolyPhen, one was predicted damaging by PolyPhen, and one was a nonsense change, a protein truncating mutation type not scored by either program (Dataset EV3). Of the three variants located in the death effector domains, two were predicted damaging by both programs. In our data, we found seven non-synonymous and one splice site change in SMARCB1 (Fig 3, Dataset EV3). Of the seven non-synonymous changes, four—including a hot spot mutation in codon 377—were predicted to be damaging by both SIFT and PolyPhen, and two were predicted to be damaging by SIFT. Again, for the nonsense and splice site change, no prediction was made. Of the seven non-synonymous variants, three were found within the functional domain of the gene (Pfam domains, Fig 3). STK38L has

been shown to promote cell survival and invasion in MSS CRC cell lines (Suzuki *et al*, 2006).

*SMARCB1* in turn is a previously known tumor suppressor gene (Suzuki *et al*, 2006). In our data, we observed six non-synonymous changes and one splice site variant in *STK38L*, and five of the six non-synonymous variants—including a mutation hot spot in codon 105—are found within the protein kinase domain of the gene (Pfam domains, Fig 3, Dataset EV3; Finn *et al*, 2016). The hot spot mutation was predicted damaging by both SIFT and PolyPhen (Dataset EV3). The remaining two of the top eight genes (*CCDC47* and *ENO3*) have to our knowledge not been implicated in CRC before.

From the top eight genes, we selected *SMARCB1* and *STK38L*—which display plausible growth associated functions and to our knowledge have not been implicated in MSI CRC before—for further validation in functional studies.

### Microscopy analysis shows normal localization of mutant SMARCB1 and STK38L

Mutations typically mediate their oncogenic potential by changing protein function via three key molecular mechanisms:

Table 1.   Ranked list of genes by OncodriveFML. A summary of the ranking by OncodriveFML and MutSigCV as well as the mutation frequencies of the genes.

| Gene | ENSG | Standing from OncodriveFML | Standing from MutSigCV | Mutation frequency of the non-synonymous changes in the gene in the discovery set of 36 MSI CRCs | Mutation frequency of the non-synonymous changes in the gene in the validation set of 93 MiSeq-sequenced samples | Mutation frequency of the non-synonymous changes in the gene in the 129 samples (discovery set + validation set) |
|---|---|---|---|---|---|---|
| BRAF | ENSG00000157764 | 1 | 1 | 33.33 | 35.48 | 34.88 |
| CASP8 | ENSG00000064012 | 2 | 4 | 13.89 | 5.38 | 7.75 |
| STK38L | ENSG00000211455 | 3 | 71 | 8.33 | 3.23 | 4.65 |
| SMARCB1 | ENSG00000099956 | 4 | 21 | 11.11 | 3.23 | 5.43 |
| CCDC47 | ENSG00000108588 | 5 | 63 | 8.33 | 3.23 | 4.65 |
| PIK3CA | ENSG00000121879 | 6 | 11 | 25.00 | 15.05 | 17.83 |
| ENO3 | ENSG00000108515 | 7 | 53 | 8.33 | 3.23 | 4.65 |
| CTNNB1 | ENSG00000168036 | 8 | 10 | 16.67 | 10.75 | 12.40 |
| MAN1B1 | ENSG00000177239 | 9 | 66 | 11.11 | 5.38 | 6.98 |
| PLG | ENSG00000122194 | 10 | 22 | 13.89 | 5.38 | 7.75 |
| NPL | ENSG00000135838 | 11 | 24 | 8.33 | 2.15 | 3.88 |
| SLITRK4 | ENSG00000179542 | 12 | 39 | 13.89 | 8.60 | 10.08 |
| PEMT | ENSG00000133027 | 13 | 38 | 8.33 | 4.30 | 5.43 |
| EPB41L3 | ENSG00000082397 | 14 | 62 | 16.67 | 9.68 | 11.63 |
| CHRM1 | ENSG00000168539 | 15 | 43 | 11.11 | 4.30 | 6.20 |
| PNCK | ENSG00000130822 | 16 | 41 | 11.11 | 2.15 | 4.65 |
| CDKAL1 | ENSG00000145996 | 17 | 9 | 22.22 | 2.15 | 7.75 |
| FOXN3 | ENSG00000053254 | 18 | 5 | 16.67 | 3.23 | 6.98 |
| CRYBB1 | ENSG00000100122 | 19 | 14 | 13.89 | 2.15 | 5.43 |
| MSGN1 | ENSG00000151379 | 20 | 35 | 8.33 | 2.15 | 3.88 |
| GLUL | ENSG00000135821 | 21 | 49 | 8.33 | 3.23 | 4.65 |
| LDHD | ENSG00000166816 | 22 | 28 | 11.11 | 3.23 | 5.43 |
| WASF3 | ENSG00000132970 | 23 | 56 | 11.11 | 3.23 | 5.43 |
| TSLP | ENSG00000145777 | 24 | 18 | 8.33 | 1.08 | 3.10 |
| GDAP1L1 | ENSG00000124194 | 25 | 33 | 11.11 | 2.15 | 4.65 |
| SLC4A11 | ENSG00000088836 | 26 | 40 | 22.22 | 10.75 | 13.95 |
| CLVS1 | ENSG00000177182 | 27 | 37 | 8.33 | 2.15 | 3.88 |
| AMD1 | ENSG00000123505 | 28 | 61 | 8.33 | 1.08 | 3.10 |
| ITM2A | ENSG00000078596 | 29 | 23 | 8.33 | 2.15 | 3.88 |
| GPR108 | ENSG00000125734 | 30 | 25 | 11.11 | 2.15 | 4.65 |
| URI1 | ENSG00000105176 | 31 | 72 | 8.33 | 3.23 | 4.65 |
| FMR1 | ENSG00000102081 | 32 | 29 | 13.89 | 6.45 | 8.53 |
| EYA4 | ENSG00000112319 | 33 | 27 | 13.89 | 2.15 | 5.43 |
| OR1N1 | ENSG00000171505 | 34 | 44 | 8.33 | 2.15 | 3.88 |
| COL10A1 | ENSG00000123500 | 35 | 73 | 8.33 | 1.08 | 3.10 |
| CREB3L4 | ENSG00000143578 | 36 | 50 | 8.33 | 2.15 | 3.88 |
| SLC36A1 | ENSG00000123643 | 37 | 31 | 11.11 | 3.23 | 5.43 |
| FN3KRP | ENSG00000141560 | 38 | 15 | 8.33 | 1.08 | 3.10 |
| DTX1 | ENSG00000135144 | 39 | 52 | 13.89 | 7.52 | 9.30 |
| HS3ST2 | ENSG00000122254 | 40 | 45 | 11.11 | 5.38 | 6.98 |

**Table 1**  (continued)

| Gene | ENSG | Standing from OncodriveFML | Standing from MutSigCV | Mutation frequency of the non-synonymous changes in the gene in the discovery set of 36 MSI CRCs | Mutation frequency of the non-synonymous changes in the gene in the validation set of 93 MiSeq-sequenced samples | Mutation frequency of the non-synonymous changes in the gene in the 129 samples (discovery set + validation set) |
|------|------|------|------|------|------|------|
| KCNJ5 | ENSG00000120457 | 41 | 59 | 8.33 | 3.23 | 4.65 |
| ING3 | ENSG00000071243 | 42 | 46 | 8.33 | 1.08 | 3.10 |
| G3BP2 | ENSG00000138757 | 43 | 58 | 8.33 | 4.30 | 5.43 |
| CPA5 | ENSG00000158525 | 44 | 54 | 8.33 | 6.45 | 6.98 |
| TGFBR1 | ENSG00000106799 | 45 | 3 | 13.89 | 2.15 | 5.43 |
| DRD4 | ENSG00000069696 | 46 | 55 | 16.67 | 1.08 | 5.43 |
| ACTL6A | ENSG00000136518 | 47 | 57 | 8.33 | 1.08 | 3.10 |
| CDH8 | ENSG00000150394 | 48 | 69 | 13.89 | 5.38 | 7.75 |
| CMTM2 | ENSG00000140932 | 49 | 17 | 8.33 | 3.23 | 4.65 |
| TRIM39 | ENSG00000204599 | 50 | 7 | 8.33 | 1.08 | 3.10 |
| KIT | ENSG00000157404 | 51 | 47 | 16.67 | 5.38 | 8.53 |
| ASCL4 | ENSG00000187855 | 52 | 64 | 8.33 | 5.38 | 6.20 |
| EOGT | ENSG00000163378 | 53 | 60 | 8.33 | 1.08 | 3.10 |
| PAK7 | ENSG00000101349 | 54 | 65 | 13.89 | 2.15 | 5.43 |
| THRB | ENSG00000151090 | 55 | 19 | 8.33 | 2.15 | 3.88 |
| TAOK3 | ENSG00000135090 | 56 | 70 | 13.89 | 0.00 | 3.88 |
| ZNF419 | ENSG00000105136 | 57 | 34 | 11.11 | 89.25 | 67.44 |

altering localization, molecular interactions, or enzymatic activity. Therefore, immunofluorescence microscopy analysis of the SMARCB1 and STK38L wild-type and mutant proteins was performed via transient transfection in HeLa cells (Figs 4A and 5A). The SMARCB1 wild-type and R377C mutant proteins were detected predominantly in the nucleus, with some staining in the cytoplasm close to the nucleus (Fig 4A). Similar results were obtained with the mass spectrometry (MS)–microscopy approach (Liu *et al*, 2018), where chromosomal (nucleus), endosomal, and membrane contexts were distinguished. No differences in localization were detected between the SMARCB1 wild-type and R377C mutant proteins. Similarly, the immunofluorescence microscopy analysis of STK38L wild-type and R105W mutant proteins showed highly similar and uniform localization in the nucleus and cytoplasm (Fig 5A). In the molecular microscopy, slightly increased endosomal localization of the R105W mutant was detected. Overall, no obvious mutation-induced changes in localization were observed for either SMARCB1 or STK38L.

**Interactome analysis identifies high-confidence interactions for SMARCB1 and STK38L**

In order to obtain insight on the possible effects of the mutations on molecular level, a comprehensive interactome analysis was performed for SMARCB1 and STK38L (bait proteins) using both affinity purification mass spectrometry (AP-MS) (Varjosalo *et al*, 2013b) and BioID proximity labeling (Roux *et al*, 2012) analyses in Flp-In T-REx 293 cells (Figs 4B and 5B, Dataset EV5, Fig EV1). For SMARCB1, these analyses identified 72 high-confidence physical

(AP-MS) interactions and 127 high-confidence interactions (HCIs) from the BioID (functional and proximal interactions). Of the total of 199 interactions, 63 were detected with both methods. For STK38L, a total of 34 physical and 86 functional HCIs were detected, of which 25 were overlapping. The obtained average connectivity for both of the analyzed bait proteins, identified using AP-MS and BioID, matches well with the numbers from the published large-scale interactomics studies (Varjosalo *et al*, 2013a; Yadav *et al*, 2017; Liu *et al*, 2018).

**SMARCB1 mutant versus wild-type interactome analysis implicates changes in "carbon metabolism" and "metabolic reprogramming in colon cancer" pathways**

From the interactome analyses, a comprehensive interaction landscape view was constructed for SMARCB1 and the 136 HCIPs (Fig 4B, Dataset EV5). Our analyses captured 17 components of the BAF (SWI/SNF-A) and PBAF (SWI/SNF-B) complexes, whose subunits are commonly mutated in cancer (Hodges *et al*, 2016). However, SP16H was the only subunit of these complexes found to display a change—a 3.5-fold increase—in binding with the R377C mutant (Dataset EV5). The remaining SMARCB1 interactors were clustered based on their Gene Ontology Biological Processes (GO-BP) terms. This resulted in clusters in "organelle organization" (16 proteins), "ATP binding" (14), "RNA binding" (12), "DNA binding" (12), and "chromatin binding" (7). Of the 136 SMARCB1 HCIPs, 52 displayed a change in binding with the mutant protein; 49 negative and 3 positive changes were identified (Figs 4B and EV1, Dataset EV5). Interestingly, the majority of the proteins with lowered

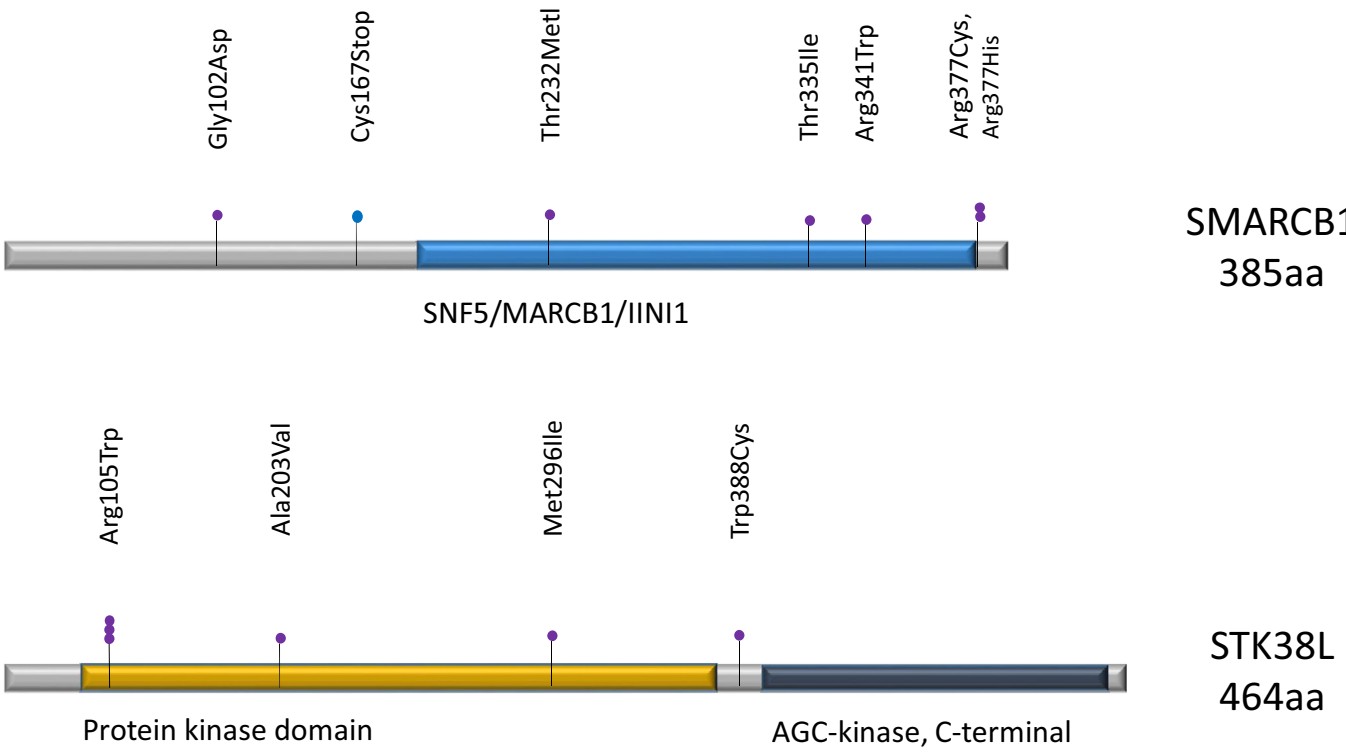

**Figure 3. Distribution of the non-synonymous somatic mutations found in *SMARCB1* and *STK38L*.**

Each sphere represents a mutation. Purple: missense; blue: nonsense.

interaction with the mutant were linked to RNA, DNA, or ATP binding. KEGG pathway mapping was then performed for the 52 proteins that displayed differential binding between the wild type and the mutant. Surprisingly, the "Carbon metabolism" pathway was identified as highly enriched (7/52 proteins linked to this pathway, Bonferroni $P$-value $4.4 \times 10^{-4}$). Similarly, the most enriched GO-BP term was the "pentose-phosphate shunt" (93.5-fold enrichment). A closer inspection of these seven proteins identified five out of seven to map to "Metabolic reprogramming in colon cancer" (Fig EV2).

**The STK38L interactome shows increased binding of the R105W mutant with 11 interactors**

As with SMARCB1, a comprehensive interactome network was constructed for STK38L and the 95 HCIPs (Fig 5B, Dataset EV5). In our network, the known STK38L interactions with the Hippo signaling pathway components MOB1B and MOB2 as well as the nuclear transport receptor Importin-11 (IPO1) were identified. However, no changes in interaction with the Hippo signaling pathway components were observed between the wild type and the R105W mutant. Additionally, we detected interactions with proteins functioning (GO-BP) in "poly(A) RNA binding" (10 HCIPs), "cell–cell adhesion" (9), "nucleic acid binding" (7), and "cytoskeleton organization" (4). Of the total of 95 HCIPs, 16 showed differential binding between the wild type and the mutant. Of these, 11 interactions (CNBP, CNOT2, HUWE1, IMA5, MAP1A, PGRC1, SCO1, SCO2, UCKL1, XPO5, and ZKSC8) increased with the mutant protein, whereas five interactions (BCR, CLIC1, SPD2B, WAC2C, and ZN569) decreased with the mutant protein (Fig 5B). As we did not detect changes in the localization of the STK38L R105W mutant, the other mechanism for the

**Figure 4. Molecular and cellular landscape of SMARCB1 wild type and R377C mutant.**

A  Immunofluorescence microscopy analysis reveals highly similar and mostly nuclear localization of the SMARCB1 wild type and the R377C mutant, visualized by anti-HA staining (green). Phalloidin and DAPI staining was used to visualize the actin cytoskeleton and the nucleus, respectively. A novel MS–microscopy approach was used to further define the molecular context of the proteins. This analysis identified possible chromosomal, endosomal, and membrane localization of the proteins. The possible endosomal localization is in agreement with the anti-HA immunofluorescence microscopy results (key: the scale bar for immunofluorescence images is 10 μm, and the color gradient on the MS–microscopy indicates the localization scores calculated by the MS–Microscopy tool)

B  The physical (AP-MS, green) and functional (BioID, red) interactions of SMARCB1 wild type and the R377C mutant (key: lower right corner). The majority of the physical interactions remained highly similar with the R377C mutant, whereas several functional interactions decreased. The interactions that decreased (< 0.6-fold) with the R377C mutant are shown with green node color, and the interactions that increased (> 2-fold) are shown in blue. The interaction map of the proteins is grouped based on participation in known protein complexes (CORUM) or on Gene Ontology Biological Processes of the proteins.

C  The cell proliferation assay in HCT116 CRC cells shows growth advantage conferred by the SMARCB1 R377C mutation compared with the wild-type SMARCB1 or the vector control. The error bars designate standard deviation (SD). Three replicates were analyzed.

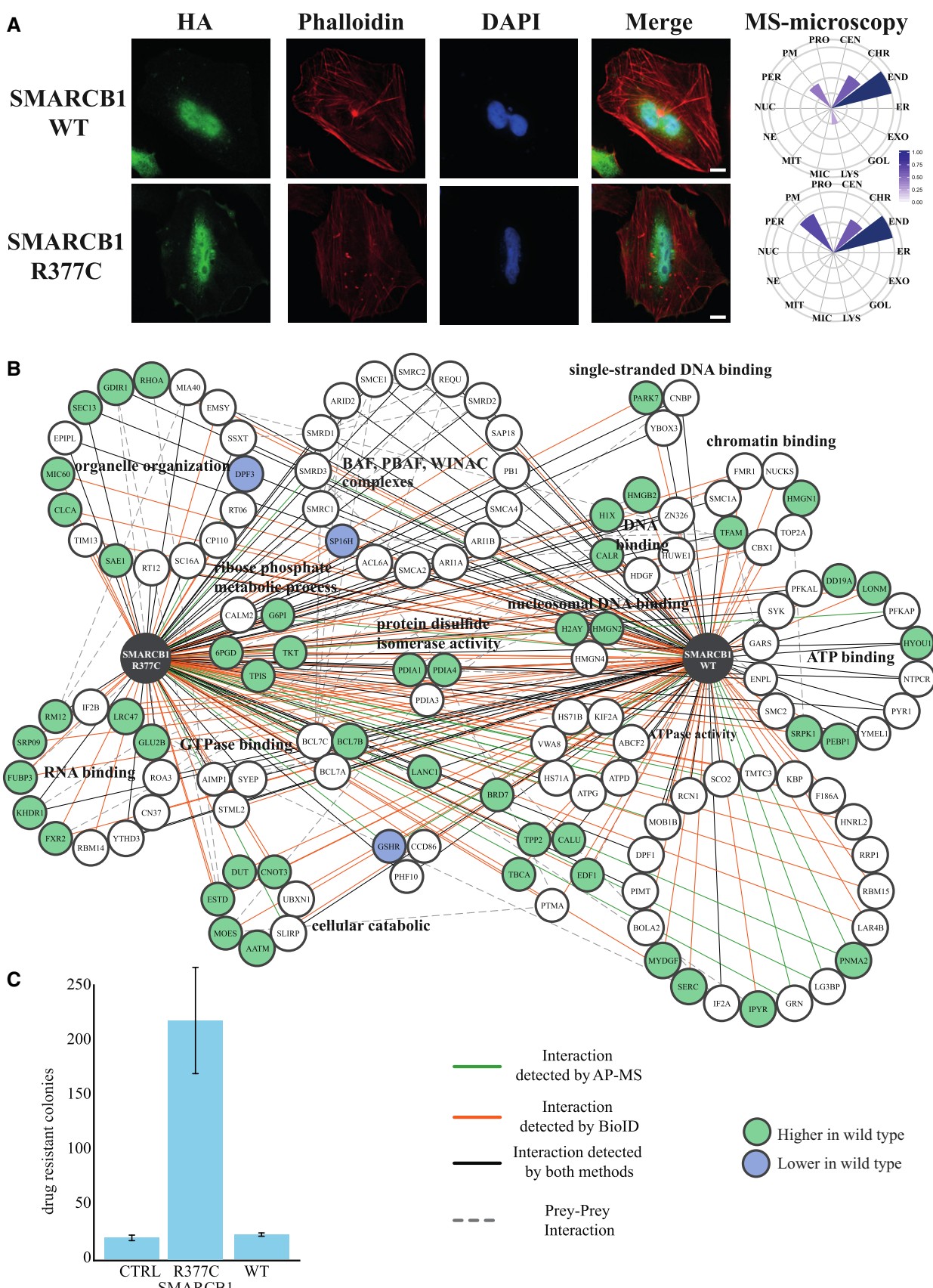

**Figure 4.**

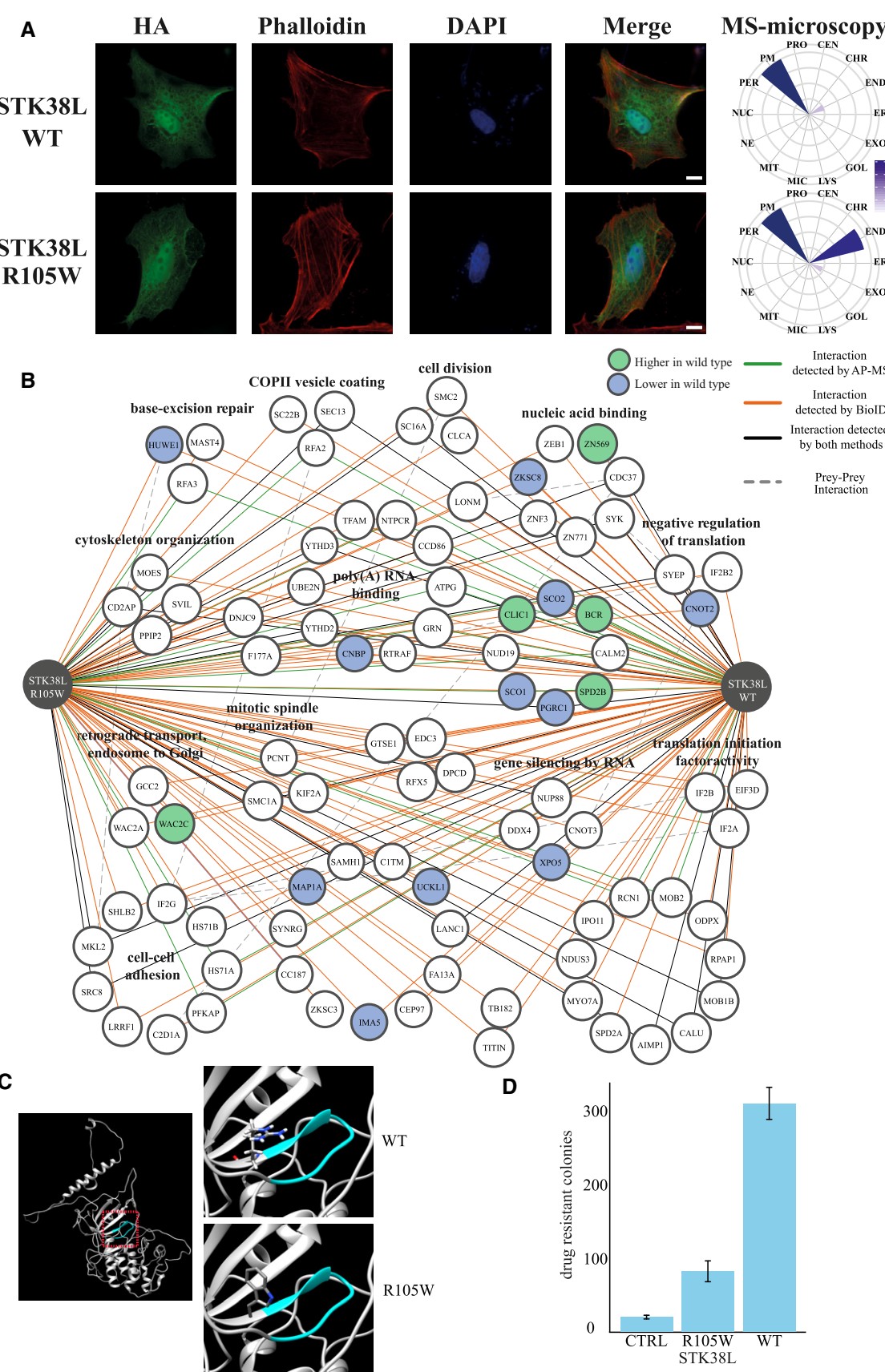

**Figure 5.**

**Figure 5.    Molecular and cellular landscape of STK38L wild type and R105W mutant.**

A    Immunofluorescence microscopy analysis of the STK38L wild type and the R105W mutant displays no clear difference in localization. Both the wild type and the R105W mutant display nuclear, cytoplasmic, and plasma membrane localization. However, the MS–microscopy analysis suggests a more prominent endosomal localization with the R105W mutant (key: The scale bar for the immunofluorescence images is 10 μm, and the color gradient on the MS–microscopy indicates the localization scores calculated by the MS-Microscopy tool).

B    The physical (AP-MS, green) and functional (BioID, red) interactions of the STK38L wild type and the R105W mutant (key: lower right corner). The R105W mutant displays increased interaction with 11 HCIPs, whereas five interactions decreased. The interactions that decreased (< 0.6-fold) with the R105W mutant are indicated with green node color, and the interactions that increased (> 2-fold) are in blue.

C    The homology modeling of the STK38L shows a shift from the positively charged arginine to a non-polar and bulky tryptophan pointing inside toward the ATP-binding site, thereby possibly inhibiting the ATP-binding and kinase activity of STK38L.

D    The cell proliferation assay in HCT116 CRC cells shows growth advantage conferred by the expression of the wild-type STK38L, whereas the R105W mutant only slightly increased the number of colonies. The error bars designate the standard deviation (SD). Three replicates were analyzed.

differential binding with interactors could be altered STK38L enzyme (kinase) activity. This is plausible as the ATP-binding region in STK38L (amino acids 96–104) is neighboring to the mutated amino acid R105. Indeed, homology modeling of STK38L shows a shift from a positively charged arginine to a non-polar tryptophan pointing inside toward the ATP-binding site, thereby possibly inhibiting the ATP-binding and kinase activity of STK38L (Fig 5C).

**The SMARCB1 R377C mutation increases colony formation in colon cancer cells**

Mutations may promote oncogenesis by altering cell cycle regulation or cell proliferation. Using the SMARCB1 and STK38L constructs stably and inducibly expressing Flp-In T-REx 293 cells, we performed a flow cytometric analysis the cell cycle with propidium iodide DNA staining. The cell cycle profiles for SMARCB1 wild type and the R377C mutant were similar to the parental cell line used as a control (Fig EV3). Similarly, the STK38L R105W mutant displayed a control-like profile. However, the STK38L wild type showed a slight increase (+5.6%) in the number of cells in G1 and a decrease (−2.8%) in the number of cells in G2 (Fig EV3). This possibly reflects a somewhat accelerated cell cycle progression.

To test whether the SMARCB1 and STK38L mutations affect CRC cell growth, we performed a cell proliferation assay in HCT116 CRC cells. With the SMARCB1 R377C mutant, we detected an increase (11.1-fold) in the number of drug-resistant colonies compared with the wild-type SMARCB1 and the vector control (Fig 4C). The wild-type SMARCB1 failed to increase colony formation (1.2-fold). Interestingly, in the case of STK38L, the wild type increased the number of drug-resistant colonies (15.7-fold), whereas the R105W mutant showed only a slight increase compared to the control (4.2-fold) (Fig 5D). The possible differences in the ability of the STK38L wild type and the R105W mutant to induce cell proliferation can be related to the differential interactomes or the possible effect of the R105W mutation on the STK38L kinase activity.

**Hot spot analysis reveals 11 genes to display additional mutations in the validation set**

Genes harboring non-synonymous or splice site hot spot changes—mutations residing in either the same or two adjacent codons, or two bases flanking an exon–intron boundary—in at least two samples were detected from the somatic point mutation data (Figs 1 and 2). Previously reported genes were omitted. Ninety hot spots from 88 genes were selected for further validation (69 hot spots in the same codon in 67 genes, and 21 hot spots in adjacent codons in

21 genes) in the extended set of 93 MSI CRCs (Dataset EV6). Two of the genes (*ALG1* and *SASH1*) contained two hot spots. Six of the selected hot spot-containing genes (*CCDC47, ENO3, LDHD, RER1, SLC4A11,* and *TMEM80*) were also found in the top 73 of the MutSigCV ranking. Also, six hot spot-containing genes Sanger-sequenced in our previous efforts (*BRAF, CMTM2, CRYBB1, CTNNB1, PIK3CA,* and *SLC36A1*) (Gylfe *et al*, 2013; Tuupanen *et al*, 2014)—harboring altogether seven hot spots—were ranked within the top 73 genes by MutSigCV and therefore qualified for validation. They were thus added to our set of hot spots. The final set entering validation therefore consisted of 97 hot spots from 94 genes (Dataset EV6). The two most commonly mutated amino acids were arginine and alanine (Appendix Fig S5A).

Of the 94 genes, 11 were found to contain hot spot mutations also in the MiSeq data: *BRAF, CORIN, CTNNB1, KLHL6, PCDHB16, PIK3CA, PLEKHG1, PROS1, SLC36A1, SPP2,* and *TROAP* (Table 2, Dataset EV3). The observed count of the total mutation hot spots in the unified data of 36 samples differed significantly from the null distribution of hot spots acquired from randomizing the mutations across the exome, with all randomized counts being less than the observed count ($P = 2 \times 10^{-5}$).

In addition to the 94 genes in the hot spot set (Datasets EV6 and EV7), there were 13 genes (*ACTL6A, ASCL4, CASP8, DTX1, EPB41L3, FMR1, FOXN3, PNCK, SLITRK4, SMARCB1, STK38L, TGFBR1,* and *URI1*) that qualified for MiSeq validation as they were found in the top 73 genes of the MutSigCV ranking, and in which a hot spot was discovered only after the MiSeq validation (Fig 2, Dataset EV3). In none of these 13 genes, however, the mutation frequency of the hot spots exceeded 5%.

# Discussion

The challenge of distinguishing driver genes from passengers is pronounced in MSI tumors due to their high mutation count, yet they may provide a sensitive model system for detection of mutation and subsequent selection. Cancer-driving genes have been considered genes whose mutations increase cell growth under the microenvironmental conditions within the cell *in vivo* (Tokheim *et al*, 2016). However, mutation frequency solely does not predict causality, but rather the mutation impact and pattern should be considered (Vogelstein *et al*, 2013).

In this study, we utilized a discovery set of 36 exome- or whole-genome-sequenced MSI CRCs and respective normals to identify new driver genes in MSI CRC based on somatic point mutations of the exome kit-targeted region of the genome (Fig 1). The top 73

**Table 2. Summary of the hot spots identified.** Information on the hot spots that contained additional mutations in the MiSeq sequencing data.

| Gene | ENSG | Chromosomal Position (GRCh37) | Base | Amino Acid Change | *n* out of the 36 NGS samples | Mutation percentage in the 36 NGS samples | *n* out of the 93 MiSeq samples | Mutation percentage in the 93 MiSeq samples | Total *n* (out of 129) | Total mutation percentage | Number of mutations in the gene targeting hot spots | Percentage of all mutations in the gene targeting hot spots |
|---|---|---|---|---|---|---|---|---|---|---|---|---|
| BRAF | ENSG00000157764 | 7:140453136 | A->T | Val600Glu | 10 | 27.8 | 32 | 34.4 | 42 | 32.6 | 42/45 | 93.3 |
| CTNNB1 | ENSG00000168036 | 3:41266124 | A->G | Thr41Ala | 2 | 5.6 | 2 | 2.2 | 4 | 3.1 | 10/17 | 58.8 |
| CTNNB1 | ENSG00000168036 | 3:41266137 | C->T | Ser45Phe | 0 | 0.0 | 6 | 6.5 | 6 | 4.7 | 10/17 | 58.8 |
| CORIN | ENSG00000145244 | 4:47625753, 4:47625751, 4:4762750 | C->T, G->A, C->T | Arg792His, Arg793Cys, Arg793His | 2 | 5.6 | 1 | 1.1 | 3 | 2.3 | NA | NA |
| KLHL6 | ENSG00000172578 | 3:183225972 | G->A | Arg262Cys | 2 | 5.6 | 1 | 1.1 | 3 | 2.3 | NA | NA |
| PCDHB16 | ENSG00000196963 | 5:140564212 | C->T | Ala693Val | 2 | 5.6 | 1 | 1.1 | 3 | 2.3 | NA | NA |
| PIK3CA | ENSG00000121879 | 3:178916876 | G->A | Arg88Gln | 1 | 2.8 | 1 | 1.1 | 2 | 1.6 | 14/25 | 56.0 |
| PIK3CA | ENSG00000121879 | 3:178916890 | C->T | Arg93Trp | 0 | 0 | 2 | 2.2 | 2 | 1.6 | 14/25 | 56.0 |
| PIK3CA | ENSG00000121879 | 3:178936091, 3:178936092, 3:178936094, 3:178936095, 3:178936096 | G->A, A-C, C->A, A->G, G->T | Glu545Lys, Glu545Ala, Gln546Lys, Gln546Arg, Gln546His | 2 | 5.6 | 4 | 4.3 | 6 | 4.7 | 14/25 | 56.0 |
| PIK3CA | ENSG00000121879 | 3:178952085 | A->G | His1047Arg | 3 | 8.3 | 1 | 1.1 | 4 | 3.1 | 14/25 | 56.0 |
| PLEKHG1 | ENSG00000120278 | 6:151161856, 6:151161857 | C->T, G->A | Arg1328Cys, Arg1328His | 2 | 5.6 | 1 | 1.1 | 3 | 2.3 | NA | NA |
| PROS1 | ENSG00000184500 | 3:93605265 | A->G | Leu413Pro | 2 | 5.6 | 1 | 1.1 | 3 | 2.3 | NA | NA |
| SLC36A1 | ENSG00000123643 | 5:150844717 | G->A | Ala136Thr | 3 | 8.3 | 1 | 1.1 | 4 | 3.1 | 4/9 | 44.4 |
| SPP2 | ENSG00000072080 | 2:234959459 | T->C | Met10Thr | 2 | 5.6 | 1 | 1.1 | 3 | 2.3 | NA | NA |
| TROAP | ENSG00000135451 | 12:49722962, 12:49722965 | C->T, C->T | Arg347Trp, Arg348Cys | 2 | 5.6 | 1 | 1.1 | 3 | 2.3 | NA | NA |

genes predicted as the most likely drivers by MutSigCV were resequenced by MiSeq sequencing in a validation set of 93 additional MSI CRCs. A newly available algorithm more suitable for smaller datasets, OncodriveFML, was utilized on the somatic point mutation data from the targeted MiSeq sequencing to yield a ranking of candidate driver genes. From these, *SMARCB1* and *STK38L* were selected for further validation in additional functional studies. To our knowledge, this study represents the first effort to uncover driver point mutations in MSI CRC utilizing deep sequencing of a large set of tumors for validation.

The two most highly mutated genes were the previously well-characterized drivers *BRAF* (33% in the discovery set, 35% in the validation set) and *PIK3CA* (25% in the discovery set, 15% in the validation set) (Fearon, 2011). The mutation percentage of the *BRAF* mutation hot spot V600E (28% in the discovery set and 34% in the validation set) was in line with previous literature (Rajagopalan *et al*, 2002). The rest of the genes were mutated with lower frequency, and distinguishing the candidate driver genes from among these was a challenge where computational prediction of mutation impact was of primary importance.

On top of the OncodriveFML ranking, there were eight genes (*BRAF, CTNNB1, CASP8, CCDC47, STK38L, ENO3, PIK3CA,* and *SMARCB1*) with a *q*-value smaller than 0.1. Three of the eight genes (*BRAF, CTNNB1,* and *PIK3CA*) are previously established oncogenic drivers of CRC, and in our data, they display the typical hot spot mutations (Polakis, 1999; Davies *et al*, 2002; Velho *et al*, 2005). *CASP8* has been listed as significantly mutated in hypermutable CRCs (Cancer Genome Atlas Network, 2012) and suggested to be a CRC suppressor gene (Kim *et al*, 2003). *SMARCB1* in turn is a previously known tumor suppressor gene (Shain & Pollack, 2013) implicated in a number of malignancies (Modena *et al*, 2005; Smith *et al*, 2012; Shain & Pollack, 2013; Bishop *et al*, 2014), including CRC (Pancione *et al*, 2013; Jauhri *et al*, 2016; Wang *et al*, 2016). *STK38L* has been shown to promote cell survival and invasion in MSS CRC cell lines (Suzuki *et al*, 2006). The remaining two of the top eight genes (*CCDC47* and *ENO3*) have to our knowledge not been implicated in CRC before. From the top eight genes, *SMARCB1* and *STK38L*—which display plausible growth associated functions and to our knowledge have not been implicated in MSI CRC before—were selected for further validation in functional studies where the effect of the mutations on the localization, molecular interactions, and enzymatic activity of the proteins was investigated.

Initially, *SMARCB1* (*SWI-SNF-related matrix-associated actin-dependent regulator of chromatin subfamily B member 1*) was shown to be biallelically inactivated in malignant rhabdoid tumor cell lines (Versteege *et al*, 1998). Later, mutations and aberrant expression of *SMARCB1* have been reported in various tumor types including familial schwannomas (Hulsebos *et al*, 2007), melanomas (Stockman *et al*, 2015), and rhabdoid tumors from different locations (Eaton *et al*, 2011). Loss of expression of SMARCB1 has been reported in colorectal adenocarcinomas and has been associated with higher histological grade, larger tumor size, poor overall survival, MSI, and the *BRAF* V600E mutation (Wang *et al*, 2016). Another CRC study has reported low expression of SMARCB1 to associate with poor differentiation, liver metastasis, and poorer survival regardless of the MMR status or tumor stage (Pancione *et al*, 2013).

In our SMARCB1 interactome analysis, 136 HCIs were detected, of which 52 displayed changes between the wild type and the R377C mutant. Interestingly, the 49 interactions that decreased in the mutant showed enrichment for glycolytic and pentose-phosphate pathway (PPP) enzymes. The synthesis of glycolytic and PPP enzymes has previously been reported to be almost ubiquitously augmented in CRC cell lines (Shibuya *et al*, 2015). Additionally, ribose-5-phosphate isomerase A (RPIA), an enzyme involved in the PPP, has been shown to be significantly elevated in CRC and to stabilize β-catenin activity and promote activation of its target genes in CRC cells (Chou *et al*, 2018).

Furthermore, five of the seven glycolytic and PPP enzymes identified in our interactome analysis (GOT2, GPI, PGD, PSAT1, and TKT) were found to map to the "Metabolic reprogramming in colon cancer" pathway. Previously, inhibition of glutamic oxaloacetic transaminase 2 (GOT2) has been shown to lead to elevated levels of reactive oxygen species (ROS) and cyclin-dependent kinase inhibitor p27-mediated cell senescence in human pancreatic ductal adenocarcinoma cells (Yang *et al*, 2018). Disruption of glucose-6-phosphate isomerase (GPI), in turn, has been shown to reduce glucose consumption and suppress lactic acid secretion in the LS174T CRC cell line, resulting in reprogramming of cells to depend on oxidative

phosphorylation and mitochondrial ATP production (de Padua *et al*, 2017). Knockdown of 6-phosphogluconate dehydrogenase (PGD) of the PPP has been shown to inhibit the growth of lung cancer cells by inducing cell senescence, which was thought to occur through accumulation of growth-inhibitory glucose metabolics (Sukhatme & Chan, 2012). Overexpression of phosphoserine aminotransferase 1 (PSAT1) in the SW480 CRC cell line was shown to increase the growth rate and survival of the cells (Vie *et al*, 2008). Finally, knockdown of transketolase (TKT) has been shown to result in a decrease in the levels of the antioxidant NADPH and an increase in ROS, and to remarkably reduce cell growth in two hepatocellular carcinoma cell lines and *in vivo* (Xu *et al*, 2016).

Glucose availability is known to be a metabolic checkpoint in cell cycle progression (Jones *et al*, 2005), and the PPP has been shown to be specifically regulated during cell cycle progression in the HT29 CRC cell line, and its inhibition to slow down the progression of the cell cycle (Vizan *et al*, 2009). In agreement with these findings, we detected increased cell proliferation induced by the expression of the SMARCB1 R337C mutant in the HCT116 CRC cell line.

*STK38L* (*serine/threonine kinase 38-like*), in turn, is a member of a family of protein serine/threonine kinases involved in the control of cell division (Tamaskovic *et al*, 2003). STK38L has been shown to be involved in the regulation of cell cycle progression by stabilizing c-myc and preventing the accumulation of p21 protein levels (Cornils *et al*, 2011a,b). STK38L has been suggested to enhance the impact of its close relative, STK38, that opposes TGF-β-mediated cell cycle arrest by limiting the phosphorylating ability of TGF-β (Pot *et al*, 2013). Also, stimulation of STK38L by IGF-1 has been shown to activate ARK5, which in turn promoted cell survival and invasion in two MSS CRC cell lines (Suzuki *et al*, 2006).

In our interactome analysis for STK38L, the previously known interactions with the Hippo signaling pathways were observed (Devroe *et al*, 2004; Meng *et al*, 2016). In addition, several interactions involved in polyA RNA binding, regulation of translation, cell–cell adhesion, and cytoskeleton organization were detected. The 16 proteins that displayed differential binding between the STK38L wild-type and the R105W mutant proteins included several proteins that have been previously linked to different cancers including CRC. Of these, cytochrome c oxidase assembly protein 1 (SCO1) has been shown to be upregulated in the Caco-2, HCT116, and HT29 CRC cell lines (Barresi *et al*, 2016). SCO1 regulates the assembly of the electron transport chain-associated cytochrome c oxidase complex along with cytochrome c oxidase assembly protein 2 (SCO2), which is in turn regulated by p53 (Nath & Chan, 2016). Mutations in *TP53* have been shown to downregulate the transcription of *SCO2* thus preventing the assembly of the cytochrome c oxidase complex, therefore promoting the cells' dependency on glycolysis for energy production. Chloride intracellular channel 1 (CLIC1) has been shown to exhibit increased protein levels in several cancers including CRC (Peretti *et al*, 2015). Knockdown of CLIC1 expression has been shown to inhibit migration and invasion of cells in the LoVo CRC cell line (Wang *et al*, 2012, 2014). Lack of SPD2B, a protein encoded by *SH3 and PX domains 2B* (*SH3PXD2B*), has been shown to result in incomplete formation of podosomes and inhibited degradation of extracellular matrix in scr-transformed fibroblasts (Buschman *et al*, 2009). RhoGEF and GTPase activating protein (BCR), in turn, is one of the two genes involved in the BCR-ABL complex associated with the Philadelphia chromosome in leukemias (Rowley, 1973).

In our hot spot effort, 97 hot spots from 94 genes were selected for MiSeq validation from the discovery set. From 11 genes (*BRAF, CORIN, CTNNB1, KLHL6, PCDHB16, PIK3CA, PLEKHG1, PROS1, SLC36A1, SPP2,* and *TROAP*), additional hot spot mutations were found in the MiSeq data. *BRAF, CTNNB1,* and *PIK3CA* are previously known MSI CRC driver genes (Shitoh *et al*, 2001; Davies *et al*, 2002; Fearon, 2011). In *SLC36A1*, a hot spot mutation has been validated in our previous effort (Tuupanen *et al*, 2014). The mutation frequencies observed in this effort and our previous study were similar and are shown in Dataset EV7. Seven of the 11 genes in which additional hot spot mutations were found in the MiSeq data—*CORIN* (R792H/R793H/R793C), *KLHL6* (R262C), *PCDHB16* (A693V), *PLEKHG1* (R1328C/R1328H), *PROS1* (L413P), *SPP2* (M10T), and *TROAP* (R347W/R348C), all containing a hot spot in 3/129 tumors—are to our knowledge novel hot spot-containing genes.

Cataloguing the genetic changes underlying cancer is essential for profound understanding of cancer biology. In this effort, a *SMARCB1* mutation exhibited altered interactions with several proteins with an enrichment of alterations for the PPP. The mutation increased colony formation in CRC cells suggesting that *SMARCB1* is a novel candidate driver gene in MSI CRCs. Also, seven novel candidate oncogenes (*CORIN, KLHL6, PCDHB16, PLEKHG1, PROS1, SPP2,* and *TROAP)* were identified based on somatic mutation hot spots. Utilizing a discovery set larger than that in our study might enable identification of yet more candidates for MSI CRC driver genes. Also, further functional work is required to validate the significance of the candidate genes discovered in this study. Cancer genes affected by point mutations—activating hot spot mutations in particular—are attractive potential therapeutic targets, and their identification should facilitate development of personalized treatments.

# Materials and Methods

### Ethics approval

The study was approved by the National Institute for Health and Welfare (THL/151/5.05.00/2017) and the Ethics Committee of the Hospital District of Helsinki and Uusimaa. All samples were derived after either an informed consent signed by the patient or authorization from the National Supervisory Authority for Welfare and Health. The study was conducted in accordance with Declaration of Helsinki and Belmont Report.

### Patient material

The discovery set of 36 sporadic MSI CRCs and corresponding blood or healthy colon tissue samples were derived from a previously characterized population-based series of 1,044 CRCs (Dataset EV8) (Aaltonen *et al*, 1998; Salovaara *et al*, 2000). Of the 36 sporadic MSI CRCs, 24 were utilized in our previous efforts where novel candidate oncogenes were identified (Gylfe *et al*, 2013; Tuupanen *et al*, 2014), and novel candidates for MSI target genes were identified (Kondelin *et al*, 2017). DNA was extracted from whole blood or fresh frozen tissue specimens using standard methods. An additional set consisting of 93 additional MSI CRCs, of which 12 were

from patients with Lynch syndrome and the rest sporadic, was available for validation (Dataset EV8). The MSI status of the tumors had been determined previously (Aaltonen *et al*, 1998; Salovaara *et al*, 2000). All tumors fulfilled the criteria for MSI high (Boland *et al*, 1998).

### Exome sequencing of 24 MSI CRCs and corresponding normals

The coding regions of the genome were enriched with the Agilent SureSelect Human All Exon Kit v1 (Agilent, Santa Clara, CA) according to the manufacturer's instructions. Paired-end short-read sequencing was performed with Illumina Genome Analyzer II machines (Illumina, Inc, San Diego, CA) at Karolinska Institute (Huddinge, Sweden), and the Institute for Molecular Medicine Finland (FIMM) Genome and Technology Center, Finland.

The read mapping and variant calling of the exome sequencing data were conducted as in our previous studies (Gylfe *et al*, 2013; Cajuso *et al*, 2014; Tuupanen *et al*, 2014).

### Whole-genome sequencing of 12 MSI CRCs and corresponding normals

Genomic DNA libraries were prepared according to Illumina and Complete Genomics (Complete Genomics Inc., Mountain View, CA, USA) paired-end sequencing service protocol. The Illumina sequencing service was performed on the Illumina HiSeq 2000 platform with paired-end reads of 100 base pairs (bp) in length. Each normal and tumor DNA sample was sequenced to a median coverage of 40× at minimum (the Complete Genomics service package was conducted with standard coverage, i.e., 40× average coverage and 90% callable diploid loci on the human reference genome). The read mapping and variant calling of the whole-genome sequencing data were conducted as in our previous study (Katainen *et al*, 2015).

### Somatic variant calling and quality control in the exome and whole-genome data

A comparative analysis and visualization tool developed in-house (BasePlayer) (Katainen *et al*, 2017) was utilized for sequencing data analysis and visualization. The sequencing data from the discovery set of 24 tumor exomes and 12 tumor genomes were filtered against data from the respective normal samples, 92 in-house blood or normal colorectal samples from CRC patients, and 26 myometrium samples to remove germline variants and artifacts from the data. The following quality filters were used for somatic variants: (i) coverage at the variant site had to be 21 or higher (high coverage was required to minimize the amount of artifacts), and (ii) the fraction of reads supporting the mutation had to be 20% or higher. In order to study SNVs only, insertions and deletions were removed from the data (Fig 1). A bed file with the areas targeted by the exome kit was utilized to unify the exome and genome data. The resulting data therefore were a list of somatic missense, nonsense, and synonymous changes as well as the few noncoding mutations found in the region targeted by the exome kit.

The effect of the variants of interest was predicted by SIFT and PolyPhen in the Ensembl Variant Effect Predictor (https://uswest.ensembl.org/info/docs/tools/vep/index.html).

### RNA-sequencing data

Gene expression levels for MutSigCV were estimated from RNA-sequencing data (Ongen *et al*, 2014). RNA from the normal samples of 22 CRC patients was extracted with RNeasy Mini Kit (Qiagen, Hilden, Germany; Appendix Table S1). The RNA-sequencing procedure is described in Ongen *et al* (2014). All RNA-seq data were processed using the Anduril software (version 1.2.21; http://csbi.ltd k.helsinki.fi) and the reference genome GRCh37 (Ovaska *et al*, 2010). The initial quality control and adapter trimming of the read data was performed with Trimmomatic (version 0.20; http://www. usadellab.org/cms/?page = trimmomatic). The trimmed data were then aligned with TopHat (version 2.0.8b; http://ccb.jhu.edu/sof tware/tophat/). The gene and isoform abundances were estimated based on Cufflinks (version 2.1.1; http://cole-trapnell-lab.github.io/ cufflinks/).

### MutSigCV

MutSigCV (v1.4) was used to rank mutated genes based on significance (Lawrence *et al*, 2013). MutSigCV analyzes SNVs discovered in DNA sequencing to identify genes that were mutated more often than expected by chance given the background mutation processes. The analyzed SNVs were annotated with Annovar (2014 Jul 04) using GRCh37 as the reference genome (Wang *et al*, 2010). Default parameters for MutSigCV were used, except for the sequencing coverage and the gene expression covariate (see below). Mutation effects were defined as noncoding, nonsilent, or silent (Appendix Table S2). The mutation effect refers to a broad class of effects that the mutation exerts on the gene. A nonsilent effect changes the protein sequence or a splice site, a silent effect is a synonymous change, and a noncoding effect is intronic or in a flanking noncoding region.

### MiSeq sequencing of the validation set of 93 MSI CRCs

From the ranking of genes derived from MutSigCV, genes with mutations in less than three tumors were left out. The coding regions of the top 73 genes (as this was feasible with the size of the MiSeq experiment) from the ranking by MutSigCV were selected for further validation with MiSeq sequencing in the validation set of 93 MSI CRCs. Also, 97 hot spots from 94 genes were selected for further validation.

Sequencing libraries were prepared with the TruSeq Custom Amplicon Index Kit (Illumina) and the TruSeq Custom Amplicon Kit v1.5 (Illumina) at Functional Genomics Unit (FuGU), Biomedicum, Helsinki. Paired-end sequencing with a read length of 150 bp was performed on Illumina MiSeq Sequencing System at FuGU. Sequence files were produced with MiSeq Control Software 2.4.1.3.

Paired-end MiSeq reads were mapped against the 1000 Genomes Project reference hs37d5 with BWA MEM (version 0.7.12) (Li, 2013). Overlapping read pair mates were clipped with the bamUtil clipOverlap tool. Regions with suspected indels were realigned with GATK IndelRealigner (GATK version 2.3-9) (Van der Auwera *et al*, 2013). Base quality scores were then normalized with GATK to produce the BAM files used in subsequent variant calling and analysis. Variants were called with GATK HaplotypeCaller with default parameters and GATK GenotypeGVCFs with default parameters

except for the minimum confidence threshold for emitting variants, which was set to 1.0 to achieve high sensitivity.

The corresponding normals were not included in the MiSeq sequencing of the validation set, but it has been shown that germline controlling with outside normal samples can exclude germline changes even more efficiently (Hiltemann *et al*, 2015).

### Somatic variant calling and quality control in the MiSeq data

The same in-house comparative analysis and visualization tool (BasePlayer) used for the exome and genome sequencing data was utilized for the MiSeq data (Katainen *et al*, 2017). The sequencing data from the validation set of 93 MSI CRCs were filtered against > 60,000 controls to remove germline variants. The controls included 213 in-house whole genomes from blood or normal colorectal samples of CRC patients, 1,092 genomes from the 1000 Genomes Project (1000 Genomes Project Consortium *et al*, 2012), 69 genomes from Complete Genomics 69 Genomes Data (http://www.complete genomics.com/public-data/69-genomes/), 402 genomes from the Kuusamo Project (Data ref: European Genome-phenome Archive EGAS00001000020, 2015), 1,941 whole genomes from the Sequencing Initiative Suomi (SiSU) project (http://www.sisuproject.fi/), 740 whole genomes from individuals from the UK10K project, and 1,692 whole genomes from twins from the UK10K project (http://www. uk10k.org/). Also 61,486 exomes (release 0.3) from individuals from Exome Aggregation Consortium (ExAC) Cambridge, MA (Lek *et al*, 2016), and 2,203 genomes from African American individuals (release 0.0.23) as well as 4,300 genomes from European Americans (release 0.0.23) from Exome Variant Server, NHLBI GO Exome Sequencing Project (ESP), Seattle, WA (http://evs.gs.washington.ed u/EVS/), were used. The control set also included 92 exomes from migraine patients as well as 14 genomes from other in-house controls. Variants with MAF $< 5 \times 10^{-5}$ were considered.

The same quality filters that were used to call somatic variants in the exome and whole-genome data were also used for the MiSeq data. The resulting data therefore were a list of somatic missense, nonsense, and synonymous changes found in < 0.10 per mil of controls.

Utilizing the COSMIC database (Forbes *et al*, 2015) and the International Cancer Genome Consortium (ICGC) Data Portal (https:// dcc.icgc.org/releases), we looked for variants in the same or adjacent codon of the variants found in our data. We only considered confirmed somatic variants in the COSMIC database and verified variants in the ICGC database. The domains of the genes were checked from Ensembl database (Finn *et al*, 2016).

### OncodriveFML

During the course of this study, a newly published algorithm, OncodriveFML, became available (Mularoni *et al*, 2016). Unlike MutSigCV, OncodriveFML is feasible on smaller datasets as it utilizes localized functional prediction of mutations rather than the background mutation rate. Hence, OncodriveFML analysis was performed on the MiSeq data from the 73 candidate genes identified by MutSigCV. OncodriveFML was run on the somatic SNV data acquired from the MiSeq sequencing of the validation set of 93 MSI CRCs. The mean of CADD scores (i.e., the default settings) was utilized. Genes with mutations in only one tumor were excluded as OncodriveFML was not able to calculate *q*-values for them.

## Expression constructs

Site-directed mutagenesis of *STK38L* and *SMARCB1* was performed to generate the recurrent mutants (*STK38L* R105W and *SMARCB1* R377C) (Data ref: Transcript: Ensembl ENST00000263121.7, 2018; Data ref: Ensembl ENST00000389032.7, 2018). The mutations for *STK38L* and *SMARCB1* were generated in gateway-compatible entry vectors obtained from a human ORFeome collection from Genome Biology Unit core facility (Research Programs Unit, HiLIFE Helsinki Institute of Life Science, Faculty of Medicine, University of Helsinki, Biocenter Finland). Site-directed mutagenesis for the three genes of interest was performed with the Q5® Site-Directed Mutagenesis Kit (New England BioLabs, Ipswich, MA, USA). The mutated clones were directly sequenced to ensure only the correct mutation was created in the inserts. The primers utilized in the mutagenesis are as follows: STK38L-F: TGGAGAGGTGtGGTTGGTCCA, STK38L-R: AAAGCTCCTC TTCCTATAACTTTCAG; and SMARCB1-F: GCGGATGAGGtGTCTTGC CAA, SMARCB1-R: CTCGTGTTCCTGTCCTGG. The three complementary DNA constructs were then cloned into C-terminal MAC-tag expression vectors with Strep, HA, and BirA tags (Liu *et al*, 2018) via a gateway LR reaction.

## Cell culture

Flp-In™ T-REx™ 293 cell lines (Invitrogen, Carlsbad, CA, USA) were cultured according to the manufacturer's instructions and utilized for generating stable cell lines that expressed the gene of interest with an inducible promoter. Cells were co-transfected with the expression vector and the pOG44 vector (Invitrogen) using the FuGENE 6 transfection reagent (Roche Applied Science, Penzberg, Germany). Four days after transfection, the cells were put in 100 μg/ml hygromycin selection media for 2 weeks. Positive clones were then pooled and amplified. Stable cell lines were each expanded to 80% confluence in 20 × 145 mm cell culture plates. Ten plates were used for the AP-MS approach and ten for the BioID experiments. Expression of the gene of interest was induced 24 h before harvesting the cells with 1 μg/ml tetracycline. For the BioID plates, 50 μM of biotin was added. Cells from five plates were pelleted as one biological sample. Therefore, each bait protein had two biological replicates in both approaches. The samples were snap-frozen and stored at −80°C.

## Interactor affinity purification

For AP-MS, the cell pellets were lysed in 3 ml of lysis buffer A (0.5% IGEPAL, 50 mM HEPES, pH 8.0, 150 mM NaCl, 50 mM NaG, 1.5 mM NaVO$_3$, 5 mM EDTA, and 0.5 mM PMSF supplemented by protease inhibitors; Sigma-Aldrich, St. Louis, MO, USA). For the BioID samples, the cell pellets were thawed in 3 ml of lysis buffer B (0.5% IGEPAL, 50 mM HEPES, pH 8.0, 150 mM NaCl, 50 mM NaF, 1.5 mM NaVO$_3$, 5 mM EDTA, 0.1% SDS, and 0.5 mM PMSF, with protease inhibitors; Sigma-Aldrich). The BioID lysates were treated with benzonase, after which they were sonicated.

The lysates were centrifuged at 16,000 *g* for 15 min, after which the supernatant was centrifuged for another 10 min to obtain cleared lysates. The lysate was then loaded consecutively on spin columns (Bio-Rad, Helsinki, Finland) containing 200 μl of Strep-Tactin beads (IBA Lifesciences, GmbH, Göttingen, Germany) prewashed with 1 ml of corresponding lysis buffer. The beads were then washed with 3 × 1 ml of lysis buffer and 4 × 1 ml of wash buffer (50 mM Tris–HCl, pH 8.0, 150 mM NaCl, 50 mM NaG, 5 mM EDTA). After the final wash, the beads were resuspended in 2 × 300 μl of elution buffer (50 mM Tris–HCl, pH 8.0, 150 mM NaCl, 50 mM NaF, 5 mM EDTA, 0.5 mM biotin) and incubated for 5 min, and eluates were collected, followed by reduction in the cysteine bonds with 5 mM Tris(2-carboxyethyl)phosphine (TCEP) for 20 min at 37°C and alkylation with 10 mM iodoacetamide (at room temperature, in the dark). Proteins were digested overnight at 37°C with sequencing-grade modified trypsin (Promega, Madison, WI, USA). After quenching with 10% TFA, the samples were desalted with C18 reverse-phase spin columns according to the manufacturer's protocol (Harvard Apparatus, Cambridge, MA, USA). The eluted samples were dried in a vacuum centrifuge and reconstituted to a final volume of 30 μl in 0.1% TFA and 1% CH$_3$CN.

## Liquid chromatography–mass spectrometry (LC-MS)

The LC-MS analysis was performed on a Q-Exactive mass spectrometer using Xcalibur version 3.0.63, coupled to an EASY-nLC 1000 system via electrospray ionization sprayer (Thermo Fisher Scientific, Waltham, MA, USA). Peptides were eluted and separated with a C18 precolumn (Acclaim PepMap 100, 75 μm × 2 cm, 3 μm, 100 Å; Thermo Fischer Scientific) and an analytical column (Acclaim PepMap RSLC, 75 μm × 15 cm, 2 μm, 100 Å; Thermo Fischer Scientific), using a 60-min buffer gradient ranging from 5 to 35% buffer B, followed by a 5-min gradient from 35 to 80% buffer B, and a 10-min gradient from 80 to 100% buffer B at the flow rate of 300 nl/min (buffer A: 0.1% formic acid in 98% HPLC-grade water and 2% acetonitrile; buffer B: 0.1% formic acid in 98% acetonitrile and 2% water). Four microliters of peptide sample was loaded from an enclosed, cooled autosampler for each sample run. Data-dependent FTMS acquisition was in positive ion mode for 80 min. A full scan (200–2,000 *m/z*) with a resolution of 70,000 was performed, followed by top 10 CID-MS2 ion trap scans with a resolution of 17,500. Dynamic exclusion was set to 30 s. The acquired MS2 spectral data files (Thermo RAW) were searched with Proteome Discoverer 1.4 (Thermo Fischer Scientific) using the SEQUEST search engine against the human component of UniProtKB/Swiss-Prot database (https://www.uniprot.org/). For the searches, trypsin was set as the digestion enzyme with a maximum of two missed cleavages permitted. Precursor mass tolerance was set to ± 15 ppm and fragment mass tolerance at 0.05 Da. Carbamidomethylation of cysteine was defined as static modification, and oxidation of methionine and biotinylation of lysine and N-termini were set as variable modifications. All reported data were based on high-confidence peptides assigned in Proteome Discoverer with FDR < 1%. For label-free quantification, spectral counts for each protein in each sample were extracted and used in relative quantification of protein abundance changes.

## Filtering and analysis of the LC-MS data

Significance Analysis of INTeractome (SAINT)-express version 3.6.065,66 (Teo *et al*, 2014) (Choi *et al*, 2011) and Contaminant Repository for Affinity Purification (CRAPome) (Mellacheruvu *et al*, 2013) were used as statistical tools for identification of specific high-confidence interactions from our AP-MS data. Sixteen GFP control runs (eight N-terminal MAC-GFP and eight C-terminal MAC-GFP

runs) were used as control counts for each hit, and the final results only considered proteins with SAINT score ≥ 0.73. This corresponds to an estimated protein-level Bayesian FDR of < 0.05. Furthermore, we used the CRAPome database with a cutoff frequency of ≥ 20% (≥ 82) except for the average spectral count fold change, which was set to ≥ 3 to assign high-confidence interactors.

GO analysis of the identified prey proteins was conducted with the DAVID gene functional analysis tool (Huang da *et al*, 2009) using the GO_BP_DIRECT and GO_MF_DIRECT gene ontology sets. The *P*-values associated with each considered annotation term were < 0.01. For pathway analyses, KEGG (www.genome.jp/kegg/) and WikiPathways (https://www.wikipathways.org/index.php/WikiPathways) were used. The interaction networks were constructed with Cytoscape (Shannon *et al*, 2003), and the protein complex information from CORUM (Ruepp *et al*, 2010) was incorporated. The known prey–prey interaction data were obtained from iRefWeb (Turner *et al*, 2010). Interaction abundance change-illustrating dotplots were created with an online dotplots visualization tool (http://prohitstools.mshri.on.ca/Dotplot/Dotplot.php) (Knight *et al*, 2015).

### Mass spectrometry–microscopy (MS–microscopy)

The MS–microscopy analyses were performed with the MS–Microscopy Web tool (http://www.biocenter.helsinki.fi/bi/protein/msmic) (Liu *et al*, 2018). The results are presented in a polar plot, with the circle divided into 14 different cellular compartments: centrosome (CEN), chromatin (CHR), endoplasmic reticulum (EM), endosome (ED), exosome (EXO), Golgi (GOL), lysosome (LYS), microtubule (MIC), mitochondria (MIT), nuclear envelope (NE), nucleolus (NUC), peroxisome (PER), plasma membrane (PM), and proteasome (PRO).

### Cell cycle experiments

For the cell cycle analysis, Muse™ Cell Cycle Kit (Merck, Kenilworth, NJ, USA) was used. Flp-In™ T-REx™ 293 cell lines expressing the corresponding transgene were grown on four 10-cm plates per cell line until 50% confluency, at which point KaryoMAX Colcemid solution (Gibco, Dublin, Ireland) was added 1:100 to two plates per cell line for 6 h. Cells were harvested by trypsinization and fixed by resuspending in ice-cold 70% EtOH. The final EtOH suspension was calculated to contain 2 million cells/ml. The resuspensions were kept at −20°C overnight. For cell counting, 300 µl of cell suspension was centrifuged at 450 *g* for 5 min and washed once with 1× PBS. Three hundred microliters of cell cycle reagent was added, and the solution was incubated at room temperature for 30 min in the dark. A total of 10,000 events were measured for each sample on Guava easyCyte single sample flow cytometer (Merck). The G2-phase peak was defined using the KaryoMAX-treated samples, while the results were obtained from untreated samples.

### Immunofluorescence microscopy

HeLa cells (ATCC, LGC Standards) were transfected with vectors containing the MAC-tagged gene of interest. Bait proteins were detected with an anti-HA antibody, followed by an Alexa Fluor 488-conjugated secondary antibody. Actin was labeled with Alexa Fluor 594 phalloidin. DAPI staining was used to define the nuclei. A wide-field fluorescence microscope (Leica, Wetzlar, Germany) with a

HCX PL APO 63×/1.40–0.60 oil objective was used to image the samples. The image files were processed with the LAS X (Leica) and ImageJ software (https://imagej.nih.gov/).

### Colony formation assay

The assay was performed as described in Messerle *et al* (1994). In short, full-length WT SMARCB1 and STK38L, as well as SMARCB1 R377C and STK38L R105W, were cloned into the modified pDEST40 (3×V5 C-terminal tag) vector (Varjosalo *et al*, 2008). HCT116 colorectal cancer cells (~5 × 10⁵; ATCC) were grown in 6-well plates and transfected with 4 µg of construct or empty vector. After 24 h, the cells were trypsinized, counted, and plated in 6-cm dishes at two different densities (Rep_1: 1 × 10⁴ cells; and Rep_2: 2 × 10⁴ cells). Cells were selected in G418 (600 µg/ml) containing media for 2–3 weeks. The colonies were stained with crystal violet and counted with ColonyArea plug-in in ImageJ.

### Structural model generation

As no atomic structures were available for STK38L, a model was generated with i-TASSER (Zhang, 2008) by running the program with default parameters using STK38L sequence from UniProt (Data ref: UniProt Q9Y2H1, 2018). Out of the five output models, the one with the highest confidence score was chosen and used to illustrate the structural context of the mutation. The illustrations were made with UCSF Chimera (Pettersen *et al*, 2004).

### Statistical analysis

The mutation frequencies of the hot spots were calculated as the number of samples mutated for each hot spot. For the hot spots in which additional mutations were found in the validation set, mutation frequencies were calculated as follows: (i) for the discovery set of 36 NGS samples, (ii) for the validation set of 93 additional MSI CRCs, and (iii) for the total of 129 samples. In case the entire gene was sequenced by MiSeq (the genes found in the MutSigCV ranking based on the discovery set: *BRAF, CTNNB1, PIK3CA*, and *SLC36A1*; Fig 2), we calculated frequencies for mutations hitting any mutation hot spot within the gene (Table 2).

In order to analyze whether the observed count of hot spots significantly differed from the expected count, a permutation test was applied. The mutations were divided into 18 different types (A>C, A>G, A>T, C>A (at CpG), C>A (not at CpG), C>G (at CpG), C>G (not at CpG), C>T (at CpG), C>T (not at CpG), G>A (at CpG), G>A (not at CpG), G>C (at CpG), G>C (not at CpG), G>T (at CpG), G>T (not at CpG), T>A, T>C, and T>G) and were then randomly redistributed to the regions captured by the exome kit. After each randomization round, the number of hot spots was counted. In order to be counted as a hot spot, there had to be two or more nonsynonymous mutations in the same or adjacent codons. Randomization was repeated 100,000 times to obtain null distribution. From this distribution, the two-sided empirical *P*-value was calculated.

### Sanger sequencing

To study the validity of the hot spot mutation appearing in *ZNF419* in the MiSeq data (Dataset EV3), Sanger sequencing of the region

**The paper explained**

**Problem**

Understanding genetic components of cancer is a prerequisite of personalized treatment. Comprehensive evaluation of MSI CRC point mutations has been lacking. To date, only few genes with causative point mutations have been identified in MSI CRC. Most of these have been flagged by missense mutation hot spots, a mutation pattern typical of oncogenes.

**Results**

In this study, we systematically searched for candidate cancer-driving genes based on mutation significance in somatic mutation data from MSI CRC. *SMARCB1* exhibited enrichment of alterations in interactions for the pentose-phosphate pathway as well as increased colony formation in CRC cells, therefore emerging as our prime candidate for a novel MSI CRC driver gene. *STK38L*, in turn, exhibited altered interaction with several interaction partners, of which many have been previously linked to cancer. Also, seven novel candidate oncogenes—*CORIN*, *KLHL6*, *PCDHB16*, *PLEKHG1*, *PROS1*, *SPP2*, and *TROAP*—were identified based on somatic mutation hot spots.

**Impact**

Cancer genes affected by point mutations—activating hot spot mutations in particular—are attractive potential therapeutic targets, and their identification will facilitate development of personalized management strategies.

was performed. The change was found in a region challenging to replicate due to sequence homology. In the resulting Sanger sequences, the change was observed in both tumors and normals, and therefore was deemed not somatic.

PCR fragments were amplified with Phusion enzyme (Thermo Fisher Scientific, Waltham, MA). Purification of the PCR products was performed with the ExoSAP-IT PCR purification kit (USB Corporation, Cleveland, OH) or A'SAP PCR cleanup kit (Arctic Zymes, Tromsø, Norway), and the sequencing reactions were performed with the BigDye Terminator v.3.1 kit (Applied Biosystems) and run using 730xl DNA Analyzer (Applied Biosystems, Foster City, CA, at FIMM Genome and Technology Centre, Finland). PCR primers were designed with the Primer3 program (http://frodo.wi.mit.edu/primer3/) with GRCh37 as the reference sequence. The primer sequences are as follows: F: AAAGGCCTTACAAGTGCAGC, R: CCACTGTGAAC TTTCTGGTGT. Analysis and visualization of the sequence graphs were performed with Mutation Surveyor software (version v4.0.8; Softgenetics, State College, PA).

## Data availability

- DNA-seq data: European Genome-phenome Archive EGAS00001 003101 (https://www.ebi.ac.uk/ega/studies/EGAS00001003101)
- Protein interaction data: IMEx consortium through the IntAct database IM-26463 (https://www.ebi.ac.uk/intact/query/IM-26463).

**Expanded View** for this article is available online.

## Acknowledgements

The authors thank Sini Marttinen, Sirpa Soisalo, Marjo Rajalaakso, Inga-Lill Åberg, Iina Vuoristo, Mairi Kuris, Alison Ollikainen, and Heikki Metsola for technical assistance. This study was supported by grants from the Academy of Finland's Center of Excellence Program 2012–2017, #250345, and 2018–2025, #312041, as well as grant #294173, The Finnish Cancer Society, The Sigrid Juselius Foundation, Jane and Aatos Erkko Foundation, SYSCOL (an EU FP7 Collaborative Project), Biocentrum Finland, HiLIFE, and Instrumentarium Research Foundation. The following foundations are acknowledged for personal grants: Biomedicum Helsinki Foundation, Otto A. Malm Foundation, Ida Montin Foundation, Orion-Farmos Research Foundation, Oskar Öflunds Stiftelse, The Emil Aaltonen Foundation, and Maud Kuistila Memorial Foundation. The Doctoral Programme in Biomedicine in the Doctoral School of Health Sciences at University of Helsinki is also thanked for funding. We acknowledge the computational resources provided by the ELIXIR node, hosted at the CSC–IT Center for Science, Finland. We acknowledge Professor Aarno Palotie (FIMM and Wellcome Trust Sanger Institute, UK) and Docent Maija Wessman (FIMM and Folkhälsan Research Center, Helsinki) for providing Finnish, population-specific exome sequences for filtering. This study makes use of the data generated by the UK10K Consortium. The authors would like to thank the NHLBI GO Exome Sequencing Project and its ongoing studies which produced and provided exome variant calls for comparison: the Lung GO Sequencing Project (HL-102923), the WHI Sequencing Project (HL-102924), the Broad GO Sequencing Project (HL-102925), the Seattle GO Sequencing Project (HL-102926), and the Heart GO Sequencing Project (HL-103010). This study makes use of data generated by the UK10K Consortium, derived from samples from TWINS UK and ALSPAC cohorts. A full list of the investigators who contributed to the generation of the data is available from www.UK10K.org. Funding for UK10K was provided by the Wellcome Trust under award WT091310.

## Author contributions

JK, LS, KO, HRa, RMP, AEG, TT, SH, ST, NV, EP, and LAA contributed to planning the mutation significance analysis. JK, KS, MA, MTu, AK, MV, and LAA contributed to planning the functional studies. JK, LS, KO, HRA, RMP, JH, AEG, RK, TT, SH, ST, NV, EP, and LAA contributed to conducting the mutation significance analysis. JK, MTU, MA, and AK contributed to generation of the site-directed mutagenesis. KS, XL, LY, and MV conducted the microscopy analysis, mass spectrometry, cell proliferation, and cell cycle experiments. LR-S, AL, SK, JB, and J-PM contributed to obtaining the tumor samples. JK, AEG, TC, UAH, TT, MA, OK, PV, and ST contributed to sample preparation for sequencing. HO and ETD contributed to generation of the RNA-sequencing data. JH, KP, HRi, RK, EK, TT, MTa, JT, OK, PV, NV, and EP contributed to computational data production and management. All authors contributed to the writing of the manuscript.

## Conflict of interest

LAA has received a lecture fee from Roche Oy. The other authors declare that they have no conflict of interest.

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
