## [Review Process File · EMBO Molecular Medicine]

Comprehensive evaluation of coding region point mutations in microsatellite unstable colorectal cancer

Johanna Kondelin, Kari Salokas, Lilli Saarinen, Kristian Ovaska, Heli Rauanheimo, Roosa-Maria Plaketti, Jiri Hamberg, Xiaonan Liu, Leena Yadav, Alexandra E. Gylfe, Tatiana Cajuso, Ulrika A. Hänninen, Kimmo Palin, Heikki Ristolainen, Riku Katainen, Eevi Kaasinen, Tomas Tanskanen, Mervi Aavikko, Minna Taipale, Jussi Taipale, Laura Renkonen-Sinisalo, Anna Lepistö, Selja Koskensalo, Jan Böhm, Jukka-Pekka Mecklin, Halit Ongen, Emmanouil T. Dermitzakis, Outi Kilpivaara, Pia Vahteristo, Mikko Turunen, Sampsa Hautaniemi, Sari Tuupanen, Auli Karhu, Niko Välimäki, Markku Varjosalo, Esa Pitkänen, Lauri A. Aaltonen

Review timeline:

Submission date:	05 October 2017
Editorial Decision:	10 November 2017
Additional communication:	20 November 2017
Revision received:	04 June 2018
Editorial Decision:	03 July 2018
Revision received:	06 July 2018
Accepted:	09 July 2018

Editor: Céline Carret

Transaction Report:

1st Editorial Decision

10 November 2017

Thank you for the submission of your manuscript to EMBO Molecular Medicine and for your response to my initial inquiry. As I said, we have now heard back from the two referees whom we asked to evaluate your manuscript.

You will see from their comments that they both find the study interesting and well performed but unfortunately, both highlight the lack of functional data on the identified genes as a serious caveat precluding publication of the data in EMBO Molecular Medicine at this stage.

Should you be willing to extend your findings with functional insights, we would welcome the submission of a revised version within three months for further consideration and would like to encourage you to address all the criticisms raised as suggested to improve conclusiveness and clarity. Please note that EMBO Molecular Medicine strongly supports a single round of revision and that, as acceptance or rejection of the manuscript will depend on another round of review, your responses should be as complete as possible.

I look forward to receiving your revised manuscript.

***** Reviewer's comments *****

Referee #1 (Remarks for Author):

This manuscript describes an attempt to identify novel somatic driver genes in microsatellite unstable colon cancers. The discovery set comprised 36 tumor/normal tissue pairs. The design of the study is straightforward and the reasoning easy to follow. As expected, several novel candidate genes showed an accumulation of mutations, and in some cases the mutations were confined to small regions, hot spots. Overall, a short list of 73 candidate genes that might turn out to act as oncogenes or tumor suppressors is presented. The paper does not contain any efforts on the part of the authors to further characterize the effects downstream of their mutated genes. I have one serious question:

Tables 1 and 2 list the top candidate genes [but the number of entries is 74 rather than 73]. The ranking is straightforward with BRAF, CTNNB1, and CASP8 at the top. Mutations in these genes were much more common than in most of the remaining genes, and many of these genes are relatively unknown entities. Nevertheless the list of candidate genes is an important contribution. My question is, what happened to the genes known to be major drivers in microsatellite stable CRC: APC, KRAS, p53 etc. Not even one mutation? To me this is hard to believe; however the authors do not comment on this finding in any way. If this were my study I would at least check the most common CRC genes by Sanger sequencing of the 36 tumors.

Referee #2 (Comments on Novelty/Model System for Author):

I put novelty low here as although this study has identified potential novel driver mutations as there is no functional data of what these do (and whether they have a specific function in MMR deficient cancers), this really just acts as a resource.

Referee #2 (Remarks for Author):

This is a well-performed study aimed at finding new driver mutations in MSI colorectal cancer. Due to MMR defects these tumours carry a very high mutational burden. Following Exome sequencing of 36 tumours, the authors apply appropriate statistical approaches to find significantly mutated genes e.g. MUTSIGCV and Oncodrive. As well as mutational modelling programmes e.g. PolyPhen. They then go on to identify hotspot mutations giving further support that these mutations are likely to be functional and driving.

Although this is done well and I think would be a useful resource for people working in this area, the lack of any biological data of how these genes might cooperate makes it hard for me to see how this is appropriate for publication at Embo Mol Medicine.

There are lots of questions raised here, as these are novel driver mutations they are presumably specific to MSI cancer. Does this mean that they have a specific function here or given the mutational processes, it makes them more easy to mutate in these cancers etc etc. Therefore without some mechanistic studies on some of the new drivers identified here I feel this study will not make a significant contribution over a resource manuscript.

Additional communication between the authors and editor

20 November 2017

Authors:

Many thanks for the opportunity to provide a revised version of the work. I can see that the reviewer's are not very specific; thus your advice here would be most helpful. I propose that we

would go for the genes and experiments depicted below, and see what useful data we can derive. Your guidance here is highly appreciated; it is of course important for us to know that our efforts at least roughly match what you had in mind.

I fully understand of course that you cannot make any promises regarding acceptance etc; your simple helpful guidance is all that is desired.

We are planning to conduct further functional studies on three of our top candidate driver genes: CASP8, STK38L and SMARCB1. These genes were found among the top candidate genes in the OncodriveFML ranking (Table 2 in the manuscript).

CASP8 (caspase 8) encodes a member of the caspase family. Members of this family are centrally involved in the execution phase of cell apoptosis and have been shown to participate in tumor development. CASP8 placed fourth in a MutSig ranking of a previous study where 30 hypermutated CRCs were utilized. Another CRC study has reported mutations indicating inactivation of the apoptotic function of CASP8, and suggested it to be a CRC driver gene. Downregulation of active caspase 8 has been reported as a mechanism of acquired resistance for TRAIL treatment in MSS CRC cell lines. Also, CASP8 has been shown to be inactivated by somatic mutations in gastric carcinomas and suggested as a candidate gene for CRC susceptibility.

STK38L (serine/threonine kinase 38 like) is a member of a family of protein serine/threonine kinases involved in the control of cell division. STK38L has been shown to be involved in the regulation of cell cycle progression by stabilizing c-myc and preventing the accumulation of the p21 protein levels. Stimulation of STK38L by IGF-1 activates ARK5, which in turn has been shown to promote cell survival and invasion in two MSS colorectal cancer cell lines. To our knowledge, STK38L has not been implicated in MSI CRC before.

SMARCB1 (SWI/SNF related, matrix associated, actin dependent regulator of chromatin, subfamily b, member 1) is a tumor suppressor gene implicated in many cancers. The protein encoded by SMARCB1 is a part of a complex that relieves repressive chromatin structures, enhancing the access of the transcriptional machinery to its targets. Loss of expression of the gene has been reported in CRCs where it was associated with higher tumor grade, larger tumor size, poorer survival, and MSI.

Of these three genes, we plan to select the recurrent mutation sites for further study. Site-directed mutagenesis will be performed. The constructs will be cloned into appropriate vectors and transfected into Flp-In 293 T-Rex cells for inducible expression of the tagged genes.

We will map the possible mutation induced changes in the protein's molecular context, namely on physical and functional interactions as well as molecular localization. This will be achieved using combinatorially affinity purification and proximity labeling, coupled with quantitative mass spectrometry (MS) analysis (PMID: 23455922 and PMID: 28330616). The possible changes in the cellular localization of the wild-type and mutant protein will be assessed using fluorescence microscopy. The more detailed molecular localization will be identified by comparing the proximity labeled proteins to in-house cellular localization reference proteome map, consisting of >20 unique cellular compartments. Additionally, possible changes in the post-translational status of the proteins can be probed by post-translational modification searches of the obtained MS data.

Editor's reply:

I read your proposal with interest and discussed it with my colleagues. We feel that by performing these sets of experiments, you would aim to confirm the point mutations as loss of function, which sounds great! I believe however, that our referees hoped for more cell biology like showing that the mutants increase invasion/cell migration/3D spheres growth? Maybe you could take one of the 3 candidates and follow through in cell culture to functionally validate that these point mutations are CRC drivers indeed?

Referee #1 (Remarks for Author):

This manuscript describes an attempt to identify novel somatic driver genes in microsatellite unstable colon cancers. The discovery set comprised 36 tumor/normal tissue pairs. The design of the study is straightforward and the reasoning easy to follow. As expected, several novel candidate genes showed an accumulation of mutations, and in some cases the mutations were confined to small regions, hot spots. Overall, a short list of 73 candidate genes that might turn out to act as oncogenes or tumor suppressors is presented. The paper does not contain any efforts on the part of the authors to further characterize the effects downstream of their mutated genes. I have one serious question:

Tables 1 and 2 list the top candidate genes [but the number of entries is 74 rather than 73]. The ranking is straightforward with BRAF, CTNNB1, and CASP8 at the top. Mutations in these genes were much more common than in most of the remaining genes, and many of these genes are relatively unknown entities. Nevertheless the list of candidate genes is an important contribution

My question is, what happened to the genes known to be major drivers in microsatellite stable CRC: APC, KRAS, p53 etc. Not even one mutation? To me this is hard to believe; however the authors do not comment on this finding in any way. If this were my study I would at least check the most common CRC genes by Sanger sequencing of the 36 tumors.

We thank the reviewer for the encouraging comments, and also learn that our manuscript appears to need some polishing towards better clarity. In this study, we utilized MutSigCV to produce a ranked list of genes, from which the top 73 candidates were selected for further validation in a set of 93 additional MSI CRCs. On the MiSeq data from these 73 genes, another algorithm, OncodriveFML was run, producing a ranked list of 57 genes. In addition, we performed an analysis on mutation hot spots, for which 97 hot spots from 94 genes were selected for validation in the MiSeq sequencing of the 93 additional MSI CRCs. Eleven of these hot spot genes contained additional mutations in the validation set.

Table 1 in mentioned by the reviewer shows the final ranking of 57 genes by OncodriveFML. Table 2 in turn lists the 11 hot spot - containing genes that contained additional mutations in the MiSeq data. The combined number of entities in Tables 1 and 2 is therefore not meant to be 73, and the number of entities presented in the tables is correct. The top 73 genes from MutSigCV the reviewer refers to are listed in Appendix Table S1.

As being said, the top 73 genes from MutSigCV were selected for further validation in MiSeq sequencing. The original MutSigCV ranking, however, yielded a ranked list of 7510 genes in total and is featured in Appendix Table S1. Among these 7510 genes were APC (on place 1797), KRAS (on place 3343), and TP53 (on place 235). These genes did contain mutations in our data but were not ranked in the top 73 genes by MutSigCV and thus were not selected for the MiSeq validation. The entire workflow of the study is summarized in Figure 1. For clarity, we have added a new figure, Figure 2, to show the detailed workflow for the hot spot effort.

We have now performed further functional studies on SMARCB1 and STK38L. We performed immunofluorescence and mass spectrometry microscopy, a mass spectrometry interactome analysis, a proliferation assay, and a flow cytometric analysis of the cell cycle to study the effect of mutations in these genes on their cellular localization, molecular interactions, colony formation, and cell cycle progression, respectively. In these efforts, SMARCB1 exhibited altered interactions with several proteins with a significant enrichment of alterations for the pentose phosphate pathway. Also, a SMARCB1 mutation increased colony formation in CRC cells. STK38L in turn exhibited altered interaction with several interaction partners that have been previously linked to cancer. From these efforts, SMARCB1 emerged as our prime candidate for a novel MSI CRC driver gene.

Initially, also CASP8 was selected for further functional studies but was then left out as transfections of the plasmid were not successful. We are willing to continue our efforts on CASP8 if considered a prerequisite for publication. However, CASP8 has already been listed as a significantly

mutated gene in MSI CRC by TCGA (Cancer Genome Atlas Network. 2012) so perhaps the effort is not worth the further delay.

Referee #2 (Comments on Novelty/Model System for Author):

I put novelty low here as although this study has identified potential new driver mutations there is no functional data of what these do (and whether they have a specific function in MMR deficient cancers), this really just acts as a resource.

Referee #2 (Remarks for Author):

This is a well performed study aimed at finding new driver mutations in MSI colorectal cancer. Due to MMR defects these tumours carry a very high mutational burden. Following Exome sequencing of 36 tumours, the authors apply appropriate statistical approaches to find significantly mutated genes e.g MUTSIGCV and Oncodrive. As well as mutational modelling programme Programs eg Polyphen. They then go onto identify hotspot mutations giving further support that these mutations are likely to be functional and driving. Although this is done well and i think would be a useful resource for people working in this area, the lack of any biological data of how these genes might cooperate makes it hard for me to see how this is appropriate for publication at Embo Mol Medicine.

There are lots of questions raised here, as these are novel driver mutations they are presumably specific to MSI cancer. Does this mean that they have a specific function here or given the mutational processes, it makes them more easy to mutate in these cancers etc etc. Therefore without some mechanistic studies on some of the new drivers identified here I feel this study will not make a significant contribution over a resource manuscript.

MSI tumors carry a high mutation burden, but for this reason - tumorigenesis progresses relying on selection of point mutations rather than chromosomal instability - they may provide a sensitive model system for detection of driver mutations. We have now performed further functional studies on two of our top candidate genes. For description of these efforts, please see our reply to the first referee's comments.

In addition to the functional studies performed, the OncodriveFML results were recompiled due to changes in the program parameters. The analysis is now improved to accurately account for any overlapping genome annotation elements. This modification resulted in eight genes acquiring a qvalue smaller than 0.1 in the OncodriveFML ranking in contrast to 13 genes in our previous version of the manuscript. Also, the order of the top genes was slightly altered with no major impact on the key results. All the respective results, tables and supplementary material were updated accordingly.

Due to the additional functional studies and the improved OncodriveFML results, we have made the necessary changes in the manuscript. Highlighting all the changes made was not feasible as it made the text difficult to read. To summarize the changes we have made in the text, we have:

- added a mention on the functional studies conducted on STK38L and SMARCB1 in the last sentence of the abstract
- added a mention on SMARCB1 emerging as our top candidate gene in MSI CRC in the second last paragraph of the introduction
- added a sentence on the genes that were selected for further validation in functional studies at the end of the fifth chapter of results
- added five new chapters in results on the functional studies
- altered paragraphs 4-12 in discussion
- done minor wording to improve the fluency of the text
- moved the materials and methods paragraphs from the supplementary data into the main text, for clarity

We stated in the cover letter of the original submission that due to patient confidentiality (germ line variants can be interpreted from the data), we are authorized to distribute somatic mutation data only.

In practice, this would have to be through EGA or dbGap databases. We are delighted to commit to this. Please advise us on the matter.

2nd Editorial Decision

03 July 2018

Thank you for the submission of your revised manuscript to EMBO Molecular Medicine. We have now received the enclosed reports from the referees that were asked to re-assess it. As you will see the reviewers are now supportive and I am pleased to inform you that we will be able to accept your manuscript pending editorial final amendments.

Please submit your revised manuscript within two weeks. I look forward to seeing a revised form of your manuscript as soon as possible.

***** Reviewer's comments *****

Referee #1 (Remarks for Author):

I conclude that the manuscript is now acceptable as is.

Referee #2 (Remarks for Author):

In this paper the authors have comprehensively examined the mutations that occur in microsatellite unstable colorectal cancer. They have performed a very robust informatics analysis with confirmation by sequencing. My major concerns over the previous version of the paper were that they had not sufficient functional work to understand how these mutations may drive cancer. Obviously I accept that there is a large list of drivers here so it would be not but realistic to do them all, however the 2 that were highlighted should be examined in more depth.

To address this point the authors have done a large amount of further work looking at the functional consequences of mutations in SMARCB1 and STK38L.

Some of the work focused on proliferation and colony-forming assays to see how the mutations could cause increased proliferation/colony formation. The SMARCB1 mutation caused a profound increase in colony formation capacity. The mechanistic work focused on IP proteomics to see what the mutated proteins would bind to compared to the normal protein. The data on SMARCB1 revealed significant enriched interactions with proteins involved in cancer metabolism.

Overall I very much appreciate the work the authors have done in revision. I feel they have provided sufficient further evidence for SMARCB1 to highlight their approach has worked and this is a potential new driver in MSI CRC. Obviously there is much more to do on these mutations (E.g. comprehensive metabolomic analysis) but I feel they can be done in further studies.

2nd Revision - authors' response

06 July 2018

Authors made the requested changes.

Corresponding Author Name: Lauri A. Aaltonen

Manuscript Number: EMM-2017-08552